# Selected Physicochemical, Thermal, and Rheological Properties of Barley Starch Depending on the Type of Soil and Fertilization with Ash from Biomass Combustion

**DOI:** 10.3390/foods13010049

**Published:** 2023-12-21

**Authors:** Karolina Pycia, Ewa Szpunar-Krok, Małgorzata Szostek, Renata Pawlak, Lesław Juszczak

**Affiliations:** 1Department of Food Technology and Human Nutrition, Institute of Food Technology, College of Natural Science, University of Rzeszow, Zelwerowicza 4 St., 35-601 Rzeszow, Poland; 2Department of Plant Production, Institute of Agricultural Sciences, and Environmental Protection, College of Natural Science, University of Rzeszow, Zelwerowicza 4 St., 35-601 Rzeszow, Poland; eszpunar@ur.edu.pl; 3Department of Soil Science, Environmental Chemistry and Hydrology, College of Natural Sciences, University of Rzeszow, Zelwerowicza 8b St., 35-601 Rzeszow, Poland; mszostek@ur.edu.pl; 4RENAGRO Renata Pawlak, Poland; pawlak-renata@o2.pl; 5Department of Food Analysis and Evaluation of Food Quality, University of Agriculture in Krakow, Balicka 122, 30-149 Krakow, Poland; rrjuszcz@cyf-kr.edu.pl

**Keywords:** barley starch, fertilization, ash from biomass combustion, soil, thermal properties, rheological properties, viscosity

## Abstract

The following study analyzed the impact of fertilizing barley with fly ash from biomass combustion grown on two types of soil, Haplic Luvisol (HL) and Gleyic Chernozem (GC), on the properties of starch. The experiment was conducted in 2019 (A) and 2020 (B), and barley was fertilized with ash doses (D1–D6) differing in mineral content. In the tested barley starch samples, the amylose content, the clarity of the paste, and the content of selected minerals were determined. The thermodynamic characteristics of gelatinization and retrogradation were determined using the DSC method. Pasting characteristics, flow curves, and viscoelastic properties of starch pastes were performed. Starches differed in amylose content and paste clarity. The highest gelatinization and retrogradation enthalpy (ΔH_G_ and ΔH_R_) values were recorded for samples GCD1A and HLD5B. None of the tested factors significantly affected the pasting temperature (PT), but they had a significant impact on the remaining parameters of the pasting characteristics. The average PT value of barley starches was 90.9 °C. However, GCD2A starch had the highest maximum viscosity and the highest rheological stability during heating. GCD2A paste was characterized by the highest apparent viscosity. It was shown that all pastes showed non-Newtonian flow and shear-thinning and had a predominance of elastic features over viscous ones. The resulting gels had the characteristics of weak gels. Ash from burning wood biomass is an innovative alternative to mineral fertilizers. It was shown that the use of such soil fertilization influenced the properties of barley starch.

## 1. Introduction

Cereal cultivation is the most dominant area of agriculture both in Poland and in the world. On a global scale, crop production is dominated by the cultivation of cereals, mainly wheat, rice, and corn. In addition to those mentioned above, barley, oats, rye, and triticale are also grown. Relatively simple cultivation, rising but still moderate production costs, and multidirectional use of the obtained grain are just some of the advantages of growing cereals. The resulting grain is an excellent, biorenewable, and relatively cheap source of carbohydrates, protein, minerals, and bioactive substances important in the diets of humans and animals. It is true that wheat grain has the greatest nutritional and economic importance, but other grains, including barley grain, also have enormous and irreplaceable economic importance. Botanically, barley (*Hordeum vulgare* L.) belongs to the grasses of the Poaceae family, *Triticeae* tribe, and *Hordeum* genus. In the world production structure, it ranks fourth after wheat, rice, and corn, and in 2021/2022, its production was approximately 147.05 million tons [1]. Barley grain is usually used for animal feed, but a large part of it is used for malt production and other food purposes. It is a grain that was probably one of the first foods for humans [2]. The usable part of barley is the kernel, the chemical composition of which largely depends on the plant’s genetic conditions and treatments of agrotechnical and climatic conditions during the growing season [2,3]. The composition of barley grain is mainly dominated (80%) by polysaccharides such as starch, sugars, and fiber. The remaining ingredients are protein (from 11.5–14.2%), β-glucans (3.7–7.7%), fats (4.7–6.8%), and minerals (1.8–2.4%) [1,4]. From a nutritional point of view, an important component of barley grain, similar to oats, is β-glucans, the presence of which in the diet has a positive effect on the human organism [5]. The dominant biopolymer in the carbohydrate category is starch, which, next to cellulose, is the polymer of the greatest technological and economic importance. It is produced in the photosynthesis process in green plants and stored in amyloplasts. Starch is composed of linear amylose and branched amylopectin, the content and ratio of which determine its properties [6]. As a result, it is the dominant ingredient of most food products, and its properties determine the quality of food. It is true that potato starch is the most technologically useful. However, for economic reasons, it can be easily replaced by cereal starches, including barley starch [7].

The cultivation of cereals, including barley, depends on factors of various natures. As in the case of other cereals, the barley grain yield is influenced by the variety, type of crop rotation, soil preparation method, fertilization, and a number of factors occurring during the growing season, such as the availability of water and solar radiation and the average air temperature [8]. Nevertheless, the factors mentioned above shape not only the yield and technological quality of the grain but also its chemical composition, including the properties of starch. Examples of these relationships are studies related to the influence of conditions during the growing season [9] or the influence of variety [10] on the properties of various barley starches. From a technological point of view, the composition of the dominant grain component, i.e., starch, and its thermal and rheological properties, which may be a function of numerous biotic and abiotic factors and the interactions between them, are very important. As Liszka-Skoczylas [7] claims, in the case of potatoes, but also other crops, the functional properties of starch are mainly a function of genetic conditions, the weather during the growing season (air temperature, supply and nature of rainfall during the growing season), and agrotechnical treatments, i.e., the use of fertilizers and possible artificial irrigation. Fertilization is one of the agrotechnical factors that can be controlled and determines the grain yield. In terms of the quality of the crop, the type of fertilization used (organic or mineral), the composition of the fertilizer, the single and total dose, the frequency of fertilization, and the method of fertilizer application are important [7]. In general, fertilization may have negative connotations for many consumers. Therefore, recently, there has been an increasing interest in alternative fertilizers, such as ash from biomass combustion. This ash contains valuable elements, such as potassium, phosphorus, and magnesium, in various amounts depending on the origin of the biomass. The use of ash from biomass combustion as an addition to mineral fertilizers allows the return of valuable elements to the environment and reduces the use of mineral fertilizers rich in nitrogen [11]. 

There are numerous papers in the literature on the impact of fertilization with ash from biomass combustion on the physicochemical properties of the soil, but there are fewer reports on their impact on yielding and crop quality. Iderawumi [12] reported that ash from biomass combustion is a source of plant nutrients for most small-scale farmers growing crops in the tropics. Ash from crop residues, vegetation, cocoa husks, oil palm bunch waste, and wood increased yields of crops such as coconut, corn, okra, peanuts, oil palm, sorghum, cotton, and millet. Fertilization with ash increased the uptake of N, P, K, B, and Ca by plants. It was found that the use of ash fertilization from biomass combustion significantly increased the yield of potato tubers and their tolerance to mechanical damage under the influence of quasi-static loads [11]. In other studies [13], fertilization with biomass ash resulted in a significant increase in the yield of spring barley and Italian ryegrass and an increase in P uptake by plants, especially in soils depleted in P. Cruz-Paredes et al. [14] found that in the second growing season after the application of biomass ash, there was no effect on the yield of spring barley, while P uptake was slightly increased in the case of an application with higher doses of ash. Ash fertilization had no effect on the uptake of cadmium by aboveground plant tissues. Pycia et al. [15] showed the impact of such fertilization on the rheological and thermal properties of potato starch. However, there is a lack of information regarding such an effect on barley starch as an example of cereal starch.

Therefore, the aim of the study was to assess the impact of fertilization with ash from biomass combustion in various doses (according to the amount of K_2_O applied to the soil) on selected physicochemical, thermal, and rheological properties of barley starch from barley grain grown on two types of soil in 2019–2020.

## 2. Materials and Methods

### 2.1. Research Material

The tested material was barley starch, obtained using a laboratory method recommended for the isolation of cereal starches [10] from barley grain of the RGT Planet variety (FR, breeder RAGT 2n; brewery type) from a field experiment conducted in 2019 (A) and 2020 (B). Briefly, starch was washed from barley flour and centrifuged to remove water and possible impurities. The conditions and parameters of the field experiment are described in the Field Experiment Section. Barley grain was grown in Haplic Luvisol (HL) and Gleyic Charnozem (GC) soil and fertilized with various doses of ash from biomass combustion ranging from D2 to D6. The D1 dose was D1-NPK mineral fertilizer. Starches isolated from barley flour were obtained after grinding barley grains in a Brabender Junior mill (Duisburg, Germany), dried at a temperature of 25 ± 1 °C, ground in a laboratory mortar, and sifted through a sieve with a mesh size of 0.125 µm. Starch was washed from barley flour, and the resulting starch suspension was centrifuged to remove water and possible impurities.

#### Field Experiment

The study of the influence of fly ash from biomass combustion (BA) on changes in starch properties in spring barley grain (*Hordeum vulgare* L.) of the RGT Planet variety (FR, breeder RAGT 2n; brewery type) was carried out in 2019–2020, on a farm in Korzenica, south-eastern Poland (50°02′22″ N 22°55′20″ E). The experiment had a two-factor randomized block design with four replications and 56 plots (plot area 40.5 m^2^). The research factors were (1) soil type (Gleyic Chernozem and Haplic Luvisol), (2) varied barley fertilization: control-only N and P mineral fertilizer, D1-NPK mineral fertilizer, and D2-D6-N and P mineral fertilizer + ash from biomass. Ash doses from biomass combustion were calculated based on the potassium content, which is its most abundant component. The ash was used before sowing in the following doses (t·ha^−1^): 0.5 (D2), 1.0 (D3), 1.5 (D4), 2.0 (D5), and 2.5 (D6), which corresponded to the following doses of potassium in the form of K_2_O (kg·ha^−1^): 100, 200, 300, 400, and 500. The ash used in the experiment was created as a result of burning forest biomass (approx. 70%) and agricultural biomass (approximately 30%). The forest biomass consisted of deciduous and coniferous trees (50/50), and the agricultural biomass components were cereal straw, sunflower husk, and willow. Biomass ash was sown in autumn and then mixed with the soil during pre-winter plowing to a depth of approximately 25–30 cm. Mineral fertilization with nitrogen (N) and phosphorus (P) was constant (the same doses in all variants of the experiment). Mineral fertilizers were applied pre-sowing in the spring and mixed into the soil with a cultivator. Nitrogen was introduced into the soil in the form of RSM^®^ 32% N (aqueous solution of urea ammonium nitrate) and monoammonium phosphate (MAP) NH_4_H_2_PO_4_ (12% N-NH4), and phosphorus was introduced in the form of monoam-monium phosphate (MAP, 22.7% P) and with biomass ash (according to experimental objects D2–D6). The mineral composition of biomass ash and the amount of nutrients delivered to the soil in the experiment variants are described in detail in the article by Szpunar-Krok et al. [11]. According to CLP Regulation (EC) No. 1272/2008, the bio-mass combustion ash used in the experiment is not a hazardous substance and does not pose a threat to human health and the environment.

Before the experiment was established, the Haplic Luvisol (HL) soil was characterized by a low content of digestible forms of P, very high Mg, and average K, Fe, Mn, Zn, and Cu content. In the Gleyic Chernozem (GC) soil, the content of digestible forms of P was low, K: very low, Mg: high, and Fe, Mn, Zn and Cu: average [16]. 

Spring barley was sown on 23 March 2019 and 20 March 2020, with a density of 250 pcs·m^2^. The harvest was carried out at full maturity on 31 July 2019 and 23 July 2020. In 2019 and 2020, during the barley growing season, 300.1 mm and 355.1 mm of rainfall were recorded, respectively, and the average air temperature was 15.9 and 13.0 °C, respectively. The detailed experimental conditions, analyzed variants, and physicochemical properties of ash from biomass combustion are described by Szostek et al. [17].

### 2.2. Methods

#### 2.2.1. Selected Physicochemical Properties

Selected physicochemical parameters in barley starches were analyzed. As part of this, moisture content was determined using the thermal drying method at 130 °C, which involved measuring the weight loss after drying [18]. The amylose fraction content was determined using the spectrophotometric method, according to Morrison and Laignelet [19]. The absorbance measurement was carried out at a wavelength of λ = 630 nm, using a UV/Vis spectrophotometer ZUZI 4211/50 (DanLab, Białystok, Poland). A 1% paste was prepared from the tested starches, and its clarity [%] was tested using the spectrophotometric method [20]. For this purpose, the barley starch suspension (1 g/100 g) was heated in a water bath at a temperature of approximately 95 °C for 30 min, stirring constantly. The transmittance measurement of starch pastes cooled to room temperature at a wavelength of λ = 640 nm was performed using a UV-Vis spectrophotometer ZUZI 4211/50 (DanLab, Poland). Analyses were performed in triplicate.

#### 2.2.2. Qualitative and Quantitative Analysis of Minerals in Starches

The content of selected macro- and microelements (Ca, Mg, K, Na, Fe, Mn, Zn, and Cu) was determined by atomic absorption spectrometry (HITACHI Z-2000, Tokyo, Japan) after wet mineralization of the sample in the presence of 60% HNO_3_ [10]. In turn, the phosphorus content was determined using the spectrophotometric method [21,22]. Analyses were performed in triplicate.

#### 2.2.3. Thermodynamic Characteristics of Gelatinisation and Retrogradation by DSC

The thermodynamical characteristics of gelatinization and retrogradation were studied using a differential scanning calorimeter DSC 4000 (PerkinElmer, Waltham, MA, USA). For this purpose, a water–starch dispersion (3:1) was heated in the DSC aluminium calorimetric pan in the temperature range of 25–110 °C with a heating rate of 10 °C/min. The empty aluminium pan was used as a reference. On the basis of received thermograms, the temperatures of the onset (T_O_), peak (T_P_), and end (T_E_) of transition, ΔT (T_O_−T_E_), and the enthalpy of gelatinization (ΔH_G_) were read. After cooling, the samples were stored in a refrigerator at 4 ± 1 °C for 7 days. Retrogradation was measured by reheating the sample pans containing the examined barley starches using the same conditions as for gelatinization. The onset (T_O_), peak (T_P_), endset (T_E_) temperatures, and enthalpy (ΔH_R_) of retrogradation were read. Additionally, the index of retrogradation (%R) was calculated from the ratio of ΔH_R_ to ΔH_G_ [10,15,20].

#### 2.2.4. Pasting Characteristics by RVA

Pasting characteristics of 10% suspensions of the tested barley starches were performed using an RVA viscosity analyzer (Rapid Visco Analizer, TecMaster, Perten Instruments, Hägersten, Sweden). The samples (mixed at 160 rpm) were held at 50 °C for 1 min, then heated to 95 °C at 12 °C/min and held at 95 °C for 3 min, cooled to 50 °C at a rate of 12 °C/min, and finally, held at 50 °C for 2 min. Based on the obtained viscograms, the following parameters were read: pasting temperature (PT), maximum viscosity during heating (PV), hot paste viscosity at 95 °C (HPV), final viscosity at 50 °C (FV), breakdown value (BD), and setback value SB [10,15,20]. The assay was performed in triplicate.

#### 2.2.5. Flow and Viscosity Curves

For these experiments, barley starch pastes prepared in an RVA analyzer were used (Section 2.2.4). The rotary viscometer Rheolab QC (Anton-Paar, Graz, Austria) with a system of coaxial cylinders (cup diameter: 27.12 mm, bob diameter: 25.00 mm) was used for the determination of flow curves at a temperature of 50 °C in the shear rate range from 1 to 300 s^−1^. The experimental curves were described by the power law equation:(1)$$\eta_{ap}=K\cdot {\dot{\gamma }}^{n-1},$$
where *η_ap_* is the apparent viscosity [Pa·s], *K* is the consistency coefficient [Pa·s*^n^*], $\dot{\gamma }$ is the share rate [s^−1^], and *n* is the flow behavior index [15,20].

The curves were determined with increasing and decreasing shear rates in the range of 1 to 300 s^−1^, and the thixotropic hysteresis area was calculated.

#### 2.2.6. Mechanical Spectra of Barley Starch Gels

Determination of the mechanical spectra of barley starch gels was obtained using a Mars II (Thermo Fisher Scientific, Waltham, MA, USA) rheometer with cone/plate geometry (cone diameter: 60 mm, angle: 1, gap size: 0.052 mm) at a temperature of 25 °C [10,15,20]. The measurements were performed at a constant strain of 0.01 in the linear viscoelastic range and an angular frequency of 1–100 rad/s. The experimental curves were described by the power-law equations:*G*′ = *K*′·*ω^n^*′,(2)
*G*″ = *K*″·*ω^n^*″,(3)
where *G*′ is the storage modulus [Pa]; *G″* is the loss modulus [Pa]; *ω* is the angular frequency [rad/s]; *K*′, *K*″, *n*′, and *n*″ are constants.

#### 2.2.7. Statistical Analysis of Research Results

The results were statistically analyzed using the Statistica 13.3 program (TIBCO Software Inc., Palo Alto, CA, USA). Changes in the properties of barley starch in individual years of the experiment depending on the dose of biomass combustion ash applied were assessed with the two-way analysis of variance (ANOVA) at α = 0.05 and with the post hoc Tukey test (HDS). All analyses of starch parameters were performed in triplicate. The significance of the differences between the values of starch parameters in individual variants, obtained in 2019 and 2020, was demonstrated based on the determination of errors reproduced from individual experiments and from errors resulting from the interaction of factors with years. The estimation was made using the ANAWAR—5.3.FR calculation program (author: Prof. Franciszek Rudnicki, Poland).

## 3. Results and Discussions

### 3.1. Physicochemical Properties

As part of the analysis of selected physicochemical properties, the amylose content and clarity of 1% starch pastes were determined. The values of these parameters are shown in Table 1. The rheological properties of starch, including the gelatinization process, as well as its sensitivity to enzymes, depend on the amylose content [6,7]. The conducted two-factor analysis of variance showed, in most cases, the impact of all tested factors on the amylose content and clarity of barley starch (BS) pastes from the crops of 2019 (A) and 2020 (B). In the case of amylose content in B starches, the type of soil did not significantly affect the value of this parameter. The amylose content in starches from grain grown in 2019 and 2020 ranged from 16.4% (GCD1) to 28.2% (GCD6) and from 17.5% (HLD6) to 25.1% (HLD2), respectively. Therefore, the average amylose content in starches from the 2019 crop on GC soil was 20.1% and on HL soil 22.1%. For the 2020 crop, the average amylose content in starch from grain grown on GC soil was 20.3% and on HL soil, 20.4%. This means that the type of soil and fertilization with ashes from wood biomass combustion had an impact on the value of this parameter. However, the effect of fertilization was not clear because different amylose contents were observed depending on the increasing dose of ash, which was dominated by K, and the type of soil. Kaur et al. [6] indicate that the amylose content in starch from different potato varieties depends on genetic factors, environmental factors, and agrotechnical treatments. The same may be the case for starch from plants of other botanical origins. The research results described in the literature regarding the influence of potassium fertilization in potato cultivation on the amylose content in starch show high variability. Zhang et al. [23] observed a clear reduction in the content of amylose and phosphorus with increasing levels of K fertilization (123–405 kg/ha). This depended on the variety of potato. The cited authors explain this tendency as a decrease in the activity of starch synthase (SSS) and starch branching enzymes (SBE), the activity of which probably determines the amylose content in starch [7]. Other authors have a different opinion [6,24], noticing that K increases the activity of the SSS and SBE enzyme, which, in their opinion, is the reason for the reduced level of amylose content in the case of potato starch. However, this was not confirmed by the research on potato starch, which depends on potassium fertilization, conducted by Zhang et al. [23]. The amylose content determined in the study had similar values to those presented by Pycia et al. [10] in malting barley of various varieties. The cited authors indicated that the amylose content determined by the spectrophotemetric method ranged from 19.6% to 25.2% and depended significantly on the grain variety. In turn, Källman et al. [25] showed that the content of this fraction in barley starch varied within a wide range of 17.6%–29.3%. Källman et al. [25] and Punia [1] claim that the amylose content in barley starch is related to the size of the grains and is determined by the growing conditions, which is consistent with the observations made. The amylose content also affects the functional properties of starch because starch with a higher amylose content is more susceptible to the retrogradation process, and its pastes are more flexible [1].

The parameter related to the functional properties of starch is the clarity of pastes. The tested starches differed significantly in this parameter. It was shown that the average clarity of starch pastes from grain from the 2019 crop was 12.5% in GC soil and 14.6% in HL soil. In the case of the 2020 crop, the average clarity of pastes based on starch from grain grown in GC soil was 16.5% and in HL soil, 18.7%. In both analyzed years, pastes based on starch from grain grown in HL soil showed higher clarity compared to GC. In the case of fertilization with ash from biomass combustion, the direction of changes was not clear. In the case of GCA soil, it was shown that with increasing fertilization (D1–D6), the clarity of the pastes decreased, and in the case of HLA soil, fertilization with ash from biomass combustion resulted in an increase in clarity. The clarity of 1% barley starch pastes determined in the study was higher than that determined by Pycia et al. [10] in the case of starch from various varieties of malting barley. The average value of this parameter, according to the cited authors, was 6.9%. According to Obadi et al. [26], the average clarity value of pastes based on highland barley starch was 16.5% and was higher compared to the clarity of wheat starch paste.

The clarity of starch pastes is determined by their transparency, i.e., the degree of light transmission through the starch paste layer. The higher the clarity of the paste at a given concentration, the greater the light transmission [27]. According to the cited authors, the clarity of starch paste is positively correlated with the amylopectin content in starch. The higher the content of branched starch fraction, the higher the clarity of the paste. Other factors affecting the clarity of starch pastes include the botanical origin of starch, its chemical composition, solubility in water, the content of especially non-starch substances, and the content and ratio of amylose to amylopectin [6,26]. The clarity of starch pastes generally decreases during storage, which is related to the retrogradation process. The exception is waxy starches with a high amylopectin content, the pastes of which remain clear during refrigerated storage [27]. Obadi et al. [26] demonstrated the high clarity of highland barley waxy starch pastes. According to the cited authors, this is the result of the amylose recrystallization process resulting from the recrystallization of amylopectin. Moreover, the clarity of the paste is also influenced by the method of its preparation, i.e., the process temperature, the speed of heating the suspension, and its concentration [10,28]. According to Wang et al. [29], the amylose content in starches of various botanical origins is a function of growing conditions, such as climate, fertilization, topographic parameters of the environment, and soil type.

### 3.2. Mineral Content in Starches

The quality of cereal grain depends mainly on the genotype of the variety [30,31] but also on environmental conditions. Fertilization and soil irrigation are the main factors influencing grain yield and its chemical composition, and thus application possibilities. Among the minerals, nitrogen is a basic component of living organisms, and nitrogen fertilizer is necessary to achieve optimal grain yield and an appropriate level of protein content. Fertilizers containing N regulate amylolytic activity and the activity of enzymes regulating the process of starch biosynthesis and the size distribution of starch granules. According to Li et al. [32] and Tong et al. [33], fertilizers with high nitrogen and sulfur content cause the production of small-sized starch granules and increase their share in the grain size distribution. Fertilizing plants with high doses of nitrogen affects grain yield, protein content in granules, and the size and properties of starch granules. However, there are still little data on the impact of reduced nitrogen fertilization on the properties of cereal starch granules (gelatinization) [33]. The conducted two-factor analysis of variance indicates the significant impact of all tested factors on the content of individual macroelements (Table 2) and microelements (Table 3). The exception is copper, the content of which did not depend significantly on the tested factors. The rheological and functional properties of starch depended largely on the phosphorus content. Depending on the year of cultivation, starch granules generally did not differ in this parameter. In the case of starch A, the type of soil had no effect on the K content. It was found that HLD2B starch had the lowest P content, and GCD6A starch had the highest P content. The analyzed starches differed slightly in terms of their phosphorus content. However, starches from crops fertilized with higher doses of ash from biomass combustion, i.e., fertilized with increasing doses of potassium and phosphorus, had a higher phosphorus content compared to the control sample. This tendency does not coincide with the observations of other authors regarding the relationship between the potassium fertilization of potatoes and the P content in starch [6,23,24]. All cited authors found a decrease in the P content in starches as a result of the K fertilization of potatoes. The changes observed in the study concern barley starch fertilized with organic fertilizer, which may be the reason for the opposite tendency. In potato starch, phosphorus is bound in esters, and in barley starch, it occurs in the form of phospholipids. Nevertheless, Leonel et al. [34] claim that growing potatoes in soils with high phosphorus content resulted in an increase in the content of this element in isolated starch. Potassium (K) is an important mineral for plant growth and development. This element plays a key role in enzyme activation, metabolism, cell development, regulation of cytosolic pH, and anion–cation balance. However, plants may be exposed to stress on a global scale due to the insufficient supply of potassium in soils [11,35]. The GCD1A sample had the lowest content of this element, and HLD3B had the highest content. Ash from burning wood biomass is, on the one hand, low in nitrogen, but on the other hand, it is rich in precious potassium. The use of ash from biomass combustion as a factor supporting plant fertilization allows the return of valuable minerals to the environment and ensures soil deacidification thanks to alkaline elements. Szpunar-Krok et al. [11], examining the yield and mechanical resistance of potato tubers grown on GC and HL soil when fertilized with doses of ash from biomass combustion differing in potassium content, found that the highest yield of potato tubers and their highest mechanical strength were recorded in the case of potassium fertilization at the D3 and D4 level (188 and 282 kg·ha^−1^ K).

### 3.3. Thermodynamic Characteristics of Gelatinization and Retrogradiation by DSC of Barley Starches

The structure and nature of the granules and the physicochemical and rheological properties of starch depend on a number of factors. The most important are genetic factors, but no less important are the environmental conditions during cultivation [36,37]. Moreover, the properties of starch depend strongly on the size of the starch granules. Among the agrotechnical factors, fertilization definitely influences the size of starch granules and their size distribution. Fertilizing crops with fertilizer containing potassium and phosphorus promotes the formation of large granules. In turn, the dominance of nitrogen fertilization causes the opposite effect, i.e., the dominance of small granules. The size of starch granules affects the gelatinization temperature, which also indirectly depends on the type and amount of fertilization [7,38]. In turn, environmental factors such as temperature and the amount and distribution of precipitation are decisive in the process of starch accumulation. High temperatures during the growing season adversely affect biosynthesis and accumulation, reducing the starch content in the granules and, thus, the grain weight. In turn, the effect of high temperatures after the flowering period changes the structure of starch and the size distribution of starch granules, as well as pasting and gelatinization properties [37,39]. The parameters of the thermodynamic characteristics of gelatinization and retrogradation determined using a differential scanning calorimeter (DSC) are summarized in Table 4 and Table 5. Starch, which is heated in the presence of water, undergoes a phase transition called the gelatinization process, which generally occurs in the temperature range characteristic of a given plant species. This process can be considered as the process of melting starch crystallites [26]. The parameters of the gelatinization process are useful for assessing the culinary properties of starch. Starch that gels at a lower temperature has good cooking properties [1,9]. The size and shape of starch granules, the degree of starch crystallinity, phosphorus content and amylose chain length, and the distribution of amylopectin chain lengths usually influence the thermal properties of starch determined by the DSC method [1,40]. A higher amylopectin content in starch translates into a narrower range of gelatinization temperatures [1,41]. Characteristic phase transition temperatures (T_O_, T_P_, T_E_, and ΔH_G_) are used to evaluate the gelatinization process by DSC. These parameters depend on the architecture of the molecular structure of the crystalline regions, the ratio of starch to water, and the rate of temperature change [26]. From the point of view of the application possibilities of starch, its gelatinization properties are very important. The parameters illustrating the gelatinization process include its characteristic temperatures (T_O_, T_P_, and T_E_) and the ΔH_G_ parameter. The term gelatinization describes the process of melting the pseudocrystalline areas of amylopectin, accompanied by partial washing out of the amylose fraction and loss of microscopic birefringence, which can be imaged using polarized light microscopy. Typically, the gelatinization process is described using the modern method of differential scanning calorimetry (DSC) [42,43]. According to Xie et al. [27], gelatinization properties of starch depend on the size of starch granules, the content of non-starch substances, the formation of amylose–lipid complexes, and the structure of amylopectin [27]. The presence of amylose–lipid complexes hinders the gelatinization process. Therefore, in this case, waxy starches gelatinize more easily at a lower temperature than normal starches. This is related to the amylose content. The conducted two-factor analysis of variance of the examined parameters of the thermodynamic analysis of starch revealed different effects of the tested factors (Table 4). Only T_P_ significantly depended on all tested factors, as did the ΔH_G_ value. It was shown that in the case of the gelatinization process, the phase transformation temperatures (To, T_P_, and T_E_), ΔT, and the ΔH_G_ parameter ranged from 56.5–61.0 °C, 61.7–65.7 °C, 67.5–71.6 °C, 9.6–11.3 °C, and 5.8–11.3 J/g. The determined values of the DSC analysis parameters are similar to those presented by Li et al. [44] and the results reported for starch from bare barley grain [45]. Moreover, the DSC analysis parameters determined in this work are lower compared to the DSC parameters reported for potato starches grown under analogous experimental conditions [15] but higher compared to the parameters tested by Pycia et al. [10] for malting barley starch. The cited authors provide phase transition temperatures (T_O_, T_P_, and T_E_) in the range of 56.5–58.5 °C, 61.2–63 °C, and 67.1–68.7 °C, respectively, for starch of various varieties of malting barley. Higher values of phase transition temperatures indicate a greater demand for energy for the gelatinization process [24,44]. Punia [1] indicates the relationship between the amylose content in starch and the phase transition temperatures. The higher the amylose content, the higher the gelation temperature range and vice versa. According to Tester [9], ambient temperature influences the gelatinization properties of barley starch, and thus, this factor affects the ability of starch granules to swell. The increase in the temperature at the beginning of gelatinization as a function of temperature is associated with an increase in the temperature at the beginning of swelling. In the literature, you can find research results related to the assessment of the impact of potato fertilization on the DSC parameters of potato starches [6]. Eburneo et al. [46] showed no relationship between the N dose in soil fertilizers, the final gelatinization temperature (T_E_), and the enthalpy of the gelatinization process (ΔH_G_). However, T_O_ and T_P_ had higher values at lower doses of N fertilization. Different observations were made by Zhu et al. [47], who showed a higher enthalpy value (ΔH_G_) and lower T_P_ at higher doses of N in fertilizers. Differences in gelatinization temperatures due to fertilization with different N doses indicate that the amount of nitrogen influences the formation of amorphous and crystalline areas. A higher gelatinization temperature may guarantee the formation of more stable amorphous regions and more ordered crystal structures in the starch. The gelatinization temperature depends on the size of starch granules, which is influenced by nitrogen fertilization [7,46]. Potassium fertilization also affects the thermal properties of starch. According to Leszczyński [48] and Liszka-Skoczylas [7], an adequate supply of potassium during the growing season increases the swelling capacity of potato starch granules by approximately 14–18%, depending on the potato variety. This is probably due to the relationship between K fertilization levels, the size of starch granules, and amylose content. The presence of amylose probably inhibits the hydration of amorphous regions in starch and thus increases the gelatinization temperature. The lower water absorption and swelling capacity of the granules probably result in an increased gelatinization temperature [7,48].

DSC analysis also illustrated the retrogradation process occurring after cooling of the starch paste as a result of molecular interactions, i.e., intra and intermolecular hydrogen bonds between amylose and amylopectin chains. The retrogradation process adversely affects the texture and appearance of food products containing starch [26]. According to Obadi et al. [23], the retrogradation process of highland barley starch is less extensive compared to corn starch but more extensive than that of wheat starch. The conducted two-factor analysis of variance showed the significant impact of all tested factors, i.e., soil type, level of fertilization with ash from biomass combustion, and the interaction of these factors on the values of retrogradation process temperatures and retrogradation enthalpy ΔH_R_ in individual years of cultivation. T_O_, T_P_, T_E_, and ΔH_R_ values ranged from 38.8 °C (HLD6B) to 45.6 °C (GCD3A), from 46.8 °C (GCD2B) to 56.7 °C (GCD5A), from 58.5 °C (GCD2B) to 67.0 °C (HLD6A), and from 3.51 J/g (GCD6B) to 5.07 J/g (HLD5B), respectively. In the case of these parameters, slight statistically significant differences were observed between the experimental variants from 2019 and 2020 (Table 3). Moreover, the influence of soil type and level of fertilization with ash from biomass combustion was not clear. Pycia et al. [10], examining the influence of malting barley variety on the thermal properties of starch, showed values of phase transformation temperatures determined by the DSC method, illustrating the retrogradation process similar to those presented in this work. According to the cited authors, the variety influenced the values of these parameters.

### 3.4. Pasting Properties of Barley Starches

Pasting properties are important for selecting a product for the food or non-food industry. According to Punia [1], the gelatinization characteristics of starch depend on the amylose content, the distribution of the length of amylopectin chains, and the content of non-starch components. Figure 1 shows exemplary pasting curves determined for barley starches, and Table 6 contains the parameters of the pasting characteristics read on the basis of the course of these curves. The pasting process describes the continuous transformation of starch occurring at temperatures equal to or above the pasting temperature range. This process involves the absorption of water and the swelling of starch granules, combined with their complete disruption. This is accompanied by significant leaching of amylose. At the beginning of the pasting process, a significant increase in viscosity is observed [42,43]. The starch gelatinization process is illustrated by a number of parameters such as gelatinization temperature (PT), maximum viscosity (PV), which shows the swelling of starch before its breakdown, and BD, which is a measure of the decrease in viscosity during storage at an elevated temperature. BD describes the difference between PV and HPV and is a measure of starch granule disintegration. A low BD value indicates greater shear resistance of starch granules when measuring the pasting process [1,26]. The BD parameter can be used to measure the resistance of paste to temperature and shear stress and to assess the sensitivity of starch granules, which indicates stability in the cooking process. The viscosity of starch paste always increases during cooling. The measure of this increase is the FV parameter (final viscosity). The difference between FV and PV is the set back (SB) parameter, which indicates the ability of starch to retrograde [6,27]. SB values are also influenced by the amylose content. It is more susceptible to retrogradation than amylopectin due to its linear structure, and its long-chain structure contributes to the formation of hydrogen bonds and gel firmness [49]. According to the cited authors, maximum viscosity (PV) is a measure of the ability of starch granules to bind water and thus influences the texture of the final product. The nature of the curves (Figure 1) is characteristic of the gelatinization process of barley starch. The tested starches generally differed in terms of, for example, maximum viscosity, which was illustrated in the form of pasting characteristics parameters (Table 5). In general, all tested starches were characterized by higher FV compared to PV. Obadi et al. [26] indicate that the pasting characteristics of starch depend on the molecular weight of starch, amylose content, lipids, and the length of amylopectin chains.

It was found that the tested barley starches did not differ significantly (*p* < 0.05) in terms of pasting temperature. The average PT value for starch from grain grown in 2019 was 90.1 °C and was 1.6 °C lower compared to starches from the 2020 crop. The PT values determined in the study are slightly higher than those presented by Pycia et al. [10] for starch from various varieties of malting barley (from 86.5 to 91.3 °C) but similar to the PT values determined for starch from hulless barley (from 93.1–95.5 °C [50]. The PT value determined by the RVA method is related to the ability of starch granules to absorb water and increase their volume. The PT value is lower the faster the granules absorb water [51]. Starches of various botanical origins differ in the value of pasting temperature. The value of this parameter depends on the plant’s genetic conditions, the content of non-starch substances, mainly lipids and proteins, and the plant’s growth conditions [9,42,52,53,54]. Two-factor analysis of variance confirmed the significant statistical impact of both the types of soil, fertilization, and the interaction between factors on the value of other RVA parameters (PV, HPV, BD, FV, and SB). Despite this, it is difficult to demonstrate the clear tendency of the impact of, for example, the value of parameters illustrating the starch pasting process. Maximum viscosity is a measure of the potential of the ordered structure of starch to absorb water and swell granules [7]. GCD2A starch had the highest PV value. However, this starch also recorded the greatest decrease in viscosity (BD), i.e., it was the least stable when stirred at high temperatures. Lower BD values indicate greater resistance of starch to heating and shear stress [6,7]. These values are much higher than those presented by Pycia et al. [10]. The cited authors report that the PV of starch in various Polish malting barley varieties ranged from 133 mPa·s to 230 mPa·s. In the case of starches grown in HL A and HL B soil, it was shown that with increasing fertilization with ash from biomass combustion (D1–D5), the PV of the gruel generally decreased. This is consistent with the information provided by Liszka-Skoczylas [7] regarding the influence of K fertilization on the pasting properties of potato starch. According to the cited author, increasing K fertilization reduces the PV of starch paste, which is related to the size of starch granules, amylose content, and the phenomenon of the ionization of phosphate groups. Moreover, Kaur et al. [6] showed a positive linear correlation between PV and BD. The highest increase in viscosity during cooling (SB) was recorded by HLD5B starch. The values of the set back (SB) parameter reflect the retrogradation process of starch paste during cooling [6]. In the case of potato starches, increased potassium fertilization reduced the susceptibility of starch to retrogradation, as evidenced by reduced SB values [1,7]. This is reflected in the case of HLD5A starch (47 mPa·s). The final viscosity of FV ranged from 362 mPa·s (HLD5A) to 560 mPa·s (GCD3B). Pycia et al. [10] reported that the FV of 5% malting barley starch pastes of various varieties ranged from 251 to 411 mPa·s. The pasting characteristics of starch also depend on the supply of phosphorus during the growing season. Increased fertilization of potatoes with phosphorus has a positive effect on the PV, BD, and SB values. Therefore, the increased availability of P in the soil results in an increased maximum viscosity of the paste, which will also be less resistant to heating and more susceptible to the retrogradation process [7]. There are no data in the literature regarding the effect of ash fertilization on the properties of starch. The impact of nitrogen fertilization is much more researched. Gu et al. [55] showed that the peak viscosity (PV) is cool paste viscosity, and the breakdown viscosity (BD) of rice starch can decrease significantly with increasing levels of nitrogen applied during the growing season. In turn, Gao et al. [56] found that the accumulation of starch granules may occur during grain filling, which is influenced by the use of primarily nitrogen fertilizers. So, it is essential to improve the quality characteristics and yield of buckwheat, learn important quality characteristics, and select the appropriate dose of nitrogen fertilizers.

### 3.5. Flow Properties

The rheological properties of starch pastes are a function of many factors, including starch concentration, shear forces, and temperature measurement [27]. Starch as a food ingredient determines its quality and stability when exposed to factors such as temperature, extreme pH values, pressure, or mechanical factors, e.g., shear. The analyzed starch pastes showed the characteristics of a non-Newtonian, shear-thinning liquid, and the rheological properties of such pastes depend on the time and shear rate [10,27,57]. According to Xie et al. [27], such properties are typical for most starch pastes, and the cited authors confirm such rheological properties of highland barley-based pastes in their research. Examples of flow curves and apparent viscosity curves of pastes of the tested starches are shown in Figure 2. Barley starches differed significantly in terms of these rheological parameters. GCD2A starch paste was characterized by the highest viscosity values in the tested shear rate range (Figure 2). This was confirmed by the parameter values of the power model used to describe the experimental curves. The conducted two-factor analysis of variance confirmed the significant statistical impact of all tested factors on the values of the consistency coefficient, flow index, and hysteresis loop area. It was found that the highest value of the consistency coefficient K, indicating the initial viscosity, was the GCD2A starch paste, and the lowest was the HLD3B sample (Table 7). The starch samples differed slightly in terms of their melt flow index, although the average value of this parameter in B starches was higher than in A starches.

Starch paste should be considered a two-phase system in which the dispersing phase is a colloidal solution of amylose dissolved in water, while the dispersed phase is grain fragments composed mainly of amylopectin. The rheological properties of such a system depend on many factors, e.g., the type and concentration of starch, the presence of impurities, the measurement temperature, and the method of preparing the paste, i.e., the time and temperature of heating the suspension [51,55]. The consistency coefficient of starch may depend on the amylose content. In the case of potato starches, Pycia et al. [20] found a positive correlation between the amylose content in starches and the consistency coefficient values, as well as a negative linear correlation between the amylose content in starch and the melt flow index values [51].

It was shown that the type of soil did not have a statistically significant effect on the value of the surface area of the hysteresis loop (HA). The value of this parameter varied widely from 148 Pa s^−1^ (HLD5A) to 2012 Pa s^−1^ (HLControlB). 

### 3.6. Viscoelastic Properties of Barley Starches

Figure 3 shows examples of mechanical spectra of barley starches from grain grown in 2019 and 2020. The *G*′ and *G*″ moduli and the loss tangent (*G*″/*G*′) were also measured, which reflect the dynamic elastic nature of gels, indicating the relative measure of the associated energy loss and the energy stored per deformation cycle [26]. It was found that for all analyzed starch gels, the values of the storage modulus *G*′ were higher than the loss modulus *G*″, which clearly indicates the dominance of the elastic features over the viscous ones. Moreover, the tangent of the phase shift angle (tan δ = *G*″/*G*′) was less than unity, which also confirms the dominance of the elastic feature (Figure 4). This behavior of barley starch gel is consistent with previous observations by Pycia et al. [10]. The cited authors in all analyzed starch samples from various barley varieties also found that the values of the *G*′ modulus were higher than *G*″, which confirms the dominance of the elastic features over the viscous ones. The determined mechanical spectra were described using power law equations, the parameters of which are summarized in Table 8. The conducted two-factor analysis of variance indicates the significant impact of soil type, fertilization, and the interactions of these factors on the values of the *K*′ and *K*″ parameters, indicating the initial values of the storage modulus and loss modulus.

It was shown that the value of the *K*′ parameter ranged from 11.2 Pa s*^n^*^′^ (HLD3B) to 31.2 Pa s*^n^*^′^ (GCD2A). However, the values of the *K*″ parameter indicating the initial value of the *G*″ module ranged from 1.5 Pa s*^n^*^″^ (GCControlA) to 3.7 Pa s*^n^*^″^ (HLControlB). The marked values are similar to those presented by other authors. Pycia et al. [10] claimed that in the case of barley starches of various varieties, the values of the *K*′ parameter ranged from 19.53 Pa s*^n^*^′^ to 41.81 Pa s*^n^*^′^, and the values of the *K*″ parameter ranged from 1.77 Pa s*^n^*^″^ to 2.76 Pa s*^n^*^″^. All analyzed barley starches differed slightly in terms of the values of the parameters *n*′ and *n*″, indicating the sensitivity of the *G*′ and *G*″ moduli to changes in angular frequency. Pycia et al. [10] had similar observations.

## 4. Conclusions

In the presented work, an attempt was made to determine the impact of soil type, alternative soil fertilization with ash from wood biomass combustion, and the interaction of these factors on selected physicochemical and rheological properties of barley starch from barley cultivation in 2019 and 2020. The significant statistical impact of the tested factors on the amylose content was found, including paste clarity, gelatinization and retrogradation process parameters, pasting characteristics parameters, and viscoelastic properties. GCD6A starch had the highest amylose content, and paste based on GCD4A starch was the clearest. It was shown that the GCD2A-based paste was characterized by the highest maximum viscosity PV and the highest rheological stability during BD heating. However, GCD6B starch had the lowest parameter values. Pastes based on all starches showed the characteristics of a non-Newtonian shear-thinned fluid, with a predominance of elastic over viscous properties. The obtained experimental data expand the knowledge of the already known influence not only of the variety or climatic conditions on the properties of barley starch but, above all, describe the influence of soil type and alternative fertilization on artificial and mineral fertilizers. In the aspect of agricultural production, determining the impact of fertilization with ash from burning wood biomass is very important because it allows for conclusions regarding their impact on the properties of the product containing such starch. This also creates conditions for discussion on the possibility of replacing artificial fertilizers with cheap ash from biomass combustion, rich in phosphorus and potassium. These observations may be helpful in designing fertilizers intended for the cultivation of root crops or other economically important cereals such as wheat. The direction of the observed changes is not clear, but it allows conclusions to be drawn regarding the influence of the tested factors on the properties of barley starch.

## Figures and Tables

**Figure 1 foods-13-00049-f001:**
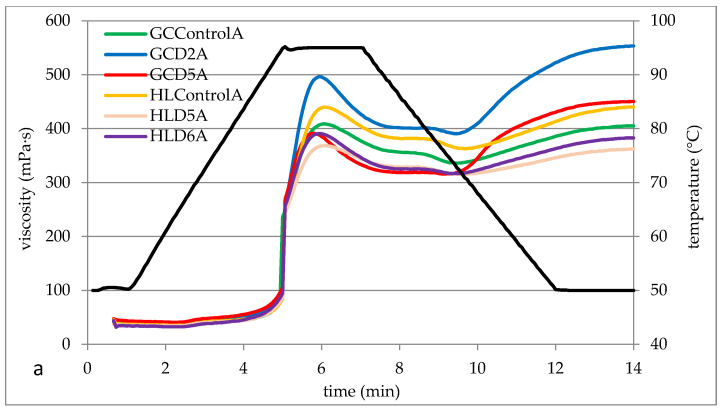
Examples of pasting curves for barley starches from the 2019 (**a**) and 2020 (**b**) crops. HL—Haplic Luvisol, GC—Gleyic Chernozem, and D1–D6—ash fertilization level.

**Figure 2 foods-13-00049-f002:**
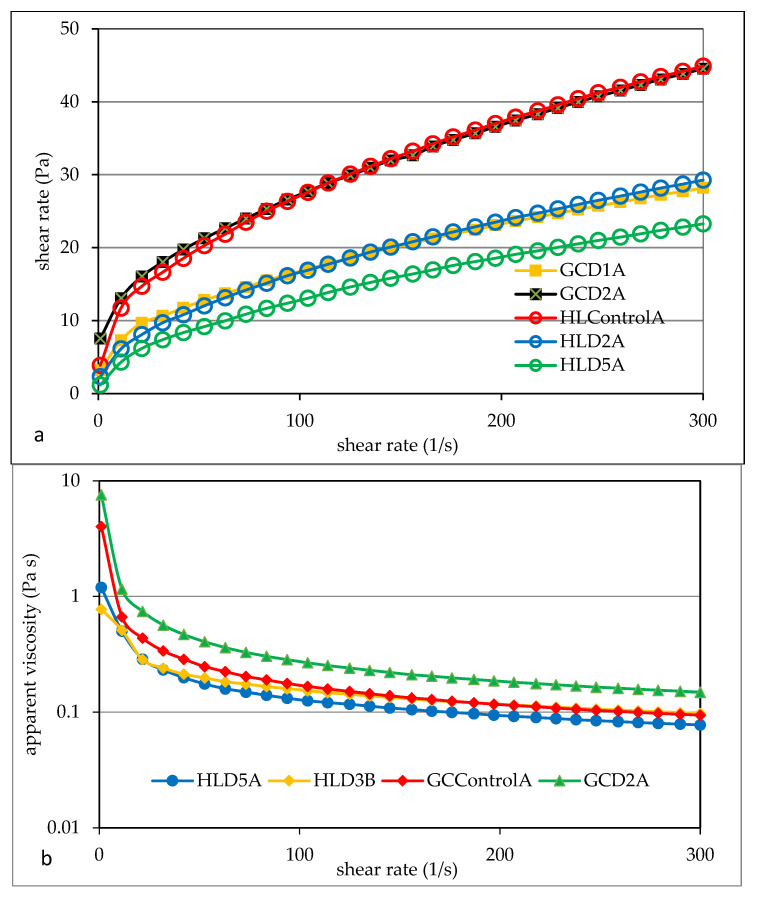
Examples of flow (**a**) and viscosity (**b**) curves of barley starches. HL—Haplic Luvisol, GC—Gleyic Chernozem, and D1–D6—ash fertilization level.

**Figure 3 foods-13-00049-f003:**
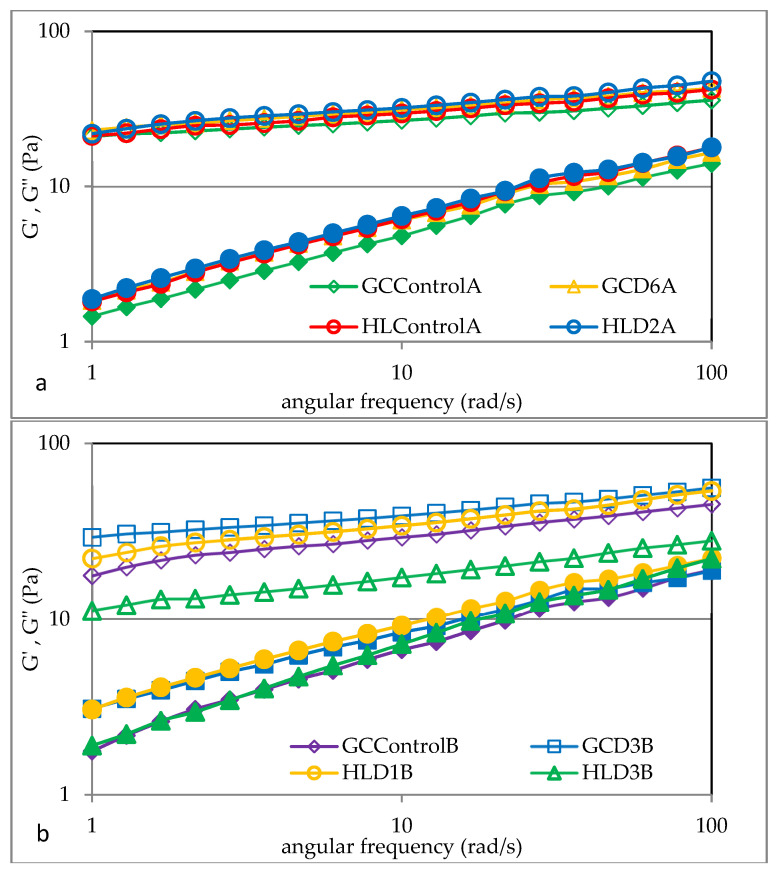
Mechanical spectra: *G*′—empty markers and *G*″—filled markers of selected barley starch gels from the crop of 2019 (**a**) and 2020 (**b**). HL—Haplic Luvisol, GC—Gleyic Chernozem, and D1–D6—ash fertilization level.

**Figure 4 foods-13-00049-f004:**
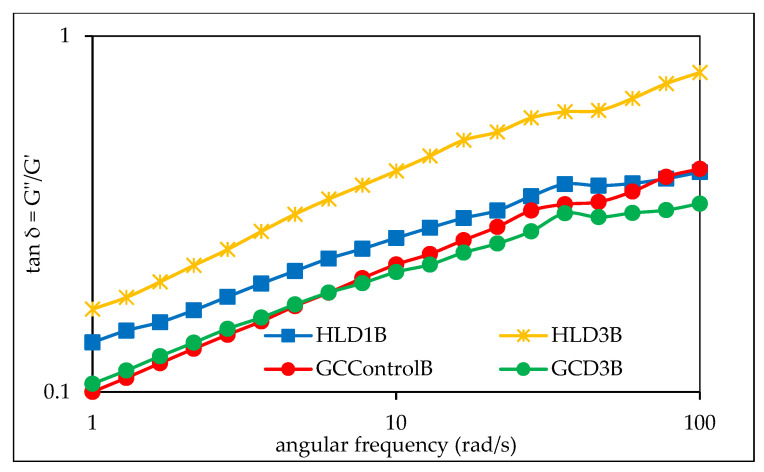
Dependence of the phase angle on angular frequency of exemplary barley starches gels. HL—Haplic Luvisol, GC—Gleyic Chernozem, and D1–D6—ash fertilization level.

**Table 1 foods-13-00049-t001:** Selected physicochemical properties of the tested barley starches.

Type of Soil	Fertilization	Amylose Content (%)	Paste Clarity (%)
2019 (A)			
Gleyic Chernozem	Control	16.9 ± 2.2 ^abA^	13.2 ± 0.56 ^deA^
	D1	16.4 ± 0.2 ^aA^	12.5 ± 0.4 ^cdA^
	D2	18.9 ± 1.1 ^abcA^	10.4 ± 0.2 ^bA^
	D3	22.0 ± 0.4 ^cdeA^	10.2 ± 0.1 ^abA^
	D4	18.1 ± 0.1 ^abA^	23.8 ± 0.4 ^hB^
	D5	20.4 ± 0.0 ^bcdA^	6.2 ± 0.1 ^abA^
	D6	28.2 ± 0.2 ^gA^	11.5 ± 0.2 ^cA^
Haplic Luvisol	Control	26.1 ± 1.5 ^fgA^	12.4 ± 0.3 ^cdA^
	D1	23.3 ± 0.4 ^defA^	14.6 ± 0.3 ^fA^
	D2	17.8 ± 1.0 ^abA^	15.8 ± 0.2 ^gA^
	D3	20.1 ± 0.8 ^bcdA^	13.9 ± 0.3 ^efA^
	D4	22.9 ± 0.6 ^defA^	14.5 ± 0.7 ^fA^
	D5	24.6 ± 0.5 ^efA^	16.5 ± 0.4 ^gA^
	D6	20.2 ± 0.2 ^bcdA^	14.6 ± 0.1 ^fA^
Two-Way ANOVA (*p*-value)			
Type of Soil		<0.001	<0.001
Fertilization		<0.001	<0.001
Type of Soil × Fertilization		<0.001	<0.001
2020 (B)			
Gleyic Chernozem	Control	24.3 ± 1.5 ^cdA^	18.9 ± 0.5 ^ghB^
	D1	18.9 ± 1.2 ^abA^	17.1 ± 0.3 ^cdB^
	D2	23.4 ± 1.5 ^cdA^	16.2 ± 0.2 ^bcB^
	D3	19.0 ± 0.0 ^abA^	15.6 ± 0.2 ^abB^
	D4	19.7 ± 0.3 ^abA^	17.2 ± 0.7 ^cdeA^
	D5	17.9 ± 1.0 ^Aa^	15.6 ± 1.2 ^abB^
	D6	19.3 ± 0.3 ^abA^	14.8 ± 0.1 ^aB^
Haplic Luvisol	Control	19.4 ± 0.5 ^abA^	17.4 ± 0.3 ^cdefB^
	D1	21.4 ± 0.7 ^bcA^	18.5 ± 0.1 ^efghB^
	D2	25.1 ± 1.4 ^dA^	18.6 ± 0.2 ^fghB^
	D3	19.2 ± 0.5 ^abA^	21.8 ± 0.6 ^iB^
	D4	21.8 ± 0.4 ^bcdA^	17.5 ± 0.2 ^cdefB^
	D5	18.6 ± 0.2 ^abA^	17.7 ± 0.4 ^defgA^
	D6	17.5 ± 0.2 ^aA^	19.7 ± 0.2 ^hB^
Two-Way ANOVA (*p*-value)			
Type of Soil		0.824	<0.001
Fertilization		<0.001	<0.001
Type of Soil × Fertilisation		<0.001	<0.001

Data are expressed as means ± SD. Values in columns marked with the same lowercase letters do not differ significantly at a significance level of 0.05. Values marked with the same capital letters indicate no significant differences between the years of the study at a significance level of 0.05. D1–D6: ash fertilization level.

**Table 2 foods-13-00049-t002:** The content of macronutrients in barley starch depending on the type of soil and fertilization with ash from biomass combustion.

Type of Soil	Fertilization	P	K	Ca	Mg	Na
		[mg kg^−1^]				
2019 (A)						
Gleyic Chernozem	Control	235.4 ± 4.36 ^abcA^	146 ± 0.4 ^bA^	69.7 ± 0.1 ^bcdA^	44.1 ± 0.0 ^abcA^	4.43 ± 0.07 ^aA^
	D1	218.0 ± 39.2 ^aA^	105 ± 1.2 ^aA^	71.8 ± 2.1 ^cdeA^	42.5 ± 3.0 ^abA^	4.90 ± 0.02 ^aA^
	D2	222.4 ± 13.1 ^abA^	109 ± 0.4 ^aA^	62.2 ± 0.5 ^abcA^	37.6 ± 0.4 ^aA^	4.79 ± 0.65 ^aA^
	D3	226.7 ± 8.7 ^abA^	131 ± 7.9 ^bA^	74.8 ± 0.2 ^defA^	43.8 ± 0.1 ^abcA^	7.36 ± 0.07 ^dA^
	D4	266.0 ± 4.4 ^bcdA^	188 ± 4.5 ^cA^	77.7 ± 2.2 ^defgA^	54.2 ± 1.6 ^deA^	6.13 ± 0.08 ^bcA^
	D5	287.8 ± 4.6 ^cdA^	172 ± 1.7 ^cA^	86.9 ± 3.4 ^ghA^	53.9 ± 1.8 ^deA^	7.27 ± 0.36 ^cdA^
	D6	292.1 ± 8.7 ^dA^	177 ± 2.6 ^cA^	88.4 ± 1.2 ^hA^	56.5 ± 2.6 ^eB^	9.46 ± 0.00 ^eB^
Haplic Luvisol	Control	244.2 ± 4.8 ^abcdA^	265 ± 1.9 ^eA^	71.8 ± 0.2 ^cdeA^	44.0 ± 1.6 ^abcA^	5.09 ± 0.23 ^abA^
	D1	252.9 ± 5.1 ^abcdA^	281 ± 4.0 ^efA^	61.0 ± 1.8 ^abA^	45.3 ± 1.7 ^bcA^	6.15 ± 0.04 ^bcA^
	D2	213.6 ± 13.1 ^aA^	293 ± 15.9 ^fA^	70.9 ± 2.2 ^bcdeA^	49.5 ± 1.9 ^cdA^	6.16 ± 0.38 ^bcA^
	D3	252.9 ± 4.9 ^abcdA^	243 ± 3.5 ^dA^	79.9 ± 1.1 ^efghA^	56.5 ± 1.5 ^eB^	5.16 ± 0.23 ^abA^
	D4	261.6 ± 5.3 ^abcdA^	239 ± 3.5 ^dA^	83.5 ± 4.6 ^fghA^	59.2 ± 0.2 ^eA^	6.94 ± 0.17 ^cdA^
	D5	257.2 ± 4.9 ^abcdA^	364 ± 2.6 ^gB^	56.9 ± 0.7 ^aA^	53.0 ± 2.4 ^deA^	6.74 ± 0.02 ^cdA^
	D6	261.6 ± 4.6 ^abcdA^	233 ± 1.0 ^dB^	90.0 ± 6.1 ^hA^	54.7 ± 0.7 ^deB^	6.88 ± 0.55 ^cdA^
Two-Way ANOVA (*p*-value)						
Type of soil		0.879	<0.001	0.021	<0.001	0.123
Fertilization		<0.001	<0.001	<0.001	<0.001	<0.001
Type of soil × Fertilization		0.009	<0.001	<0.001	<0.001	<0.001
2020 (B)						
Gleyic Chernozem	Control	235.4 ± 4.4 ^abcA^	276 ± 4 ^fgB^	69.4 ± 2.5 ^aA^	43.5 ± 1.2 ^abcA^	8.38 ± 1.13 ^deA^
	D1	257.2 ± 17.4 ^bcdeA^	223 ± 2 ^cB^	92.7 ± 1.2 ^bA^	40.3 ± 1.5 ^abA^	6.99 ± 0.09 ^abcdB^
	D2	244.2 ± 13.1 ^abcdA^	228 ± 3 ^cdB^	86.3 ± 0.1 ^bB^	44.6 ± 1.2 ^abcA^	6.86 ± 0.06 ^abcdA^
	D3	252.9 ± 8.7 ^bcdA^	272 ± 1 ^fgB^	92.7 ± 1.1 ^bB^	47.5 ± 0.6 ^cdA^	8.25 ± 0.52 ^deA^
	D4	300.8 ± 43.6 ^eA^	237 ± 11 ^cdB^	94.6 ± 3.6 ^bA^	42.8 ± 1.0 ^abcA^	7.60 ± 0.57 ^cdeA^
	D5	270.3 ± 13.1 ^cdeA^	293 ± 3 ^gB^	91.6 ± 0.4 ^bA^	48.9 ± 0.7 ^dA^	7.44 ± 0.23 ^bcdeA^
	D6	283.4 ± 8.72 ^deA^	220 ± 3 ^bcA^	87.6 ± 2.2 ^bA^	39.9 ± 0.9 ^abA^	6.85 ± 0.13 ^abcdA^
Haplic Luvisol	Control	235.4 ± 4.4 ^abcA^	270 ± 10 ^fA^	73.7 ± 5.7 ^aA^	42.9 ± 1.2 ^abcA^	9.20 ± 0.98 ^eA^
	D1	213.6 ± 4.4 ^abA^	200 ± 6 ^abA^	74.6 ± 1.1 ^aB^	38.5 ± 3.3 ^aA^	5.89 ± 0.34 ^abcA^
	D2	204.9 ± 30.5 ^aA^	367 ± 1 ^hB^	75.7 ± 0.8 ^aA^	48.9 ± 1.7 ^dA^	8.70 ± 0.36 ^deB^
	D3	257.2 ± 4.3 ^bcdeA^	376 ± 1 ^hB^	76.6 ± 1.1 ^aA^	46.6 ± 1.3 ^cdA^	8.66 ± 0.08 ^deB^
	D4	257.2 ± 4.1 ^bcdeA^	262 ± 11 ^efA^	76.1 ± 0.8 ^aA^	43.3 ± 4.8 ^abcA^	5.64 ± 0.50 ^abcA^
	D5	257.2 ± 4.0 ^bcdeA^	246 ± 2 ^deA^	74.4 ± 0.7 ^aB^	41.8 ± 1.6 ^abcA^	5.00 ± 0.27 ^aA^
	D6	261.6 ± 4.0 ^cdeA^	198 ± 1 ^aA^	76.5 ± 0.5 ^aA^	45.0 ± 0.9 ^abcA^	5.53 ± 0.14 ^abA^
Two-Way ANOVA (*p*-value)						
Type of soil		<0.001	<0.001	<0.001	0.925	0.013
Fertilization		<0.001	<0.001	<0.001	0.001	<0.001
Type of soil × Fertilization		<0.001	<0.001	<0.001	0.010	<0.001

Data are expressed as means ± SD. Values in columns marked with the same lowercase letters do not differ significantly at a significance level of 0.05. Values marked with the same capital letters indicate no significant differences between the years of the study at a significance level of 0.05. D1–D6: ash fertilization level.

**Table 3 foods-13-00049-t003:** The content of micronutrients in barley starch depending on the type of soil and fertilization with ash from biomass combustion.

Type of Soil	Fertilization	Fe	Mn	Cu	Zn
		[mg kg^−1^]			
2019 (A)					
Gleyic Chernozem	Control	3.93 ± 0.98 ^abcA^	0.54 ± 0.01 ^abA^	0.67 ± 0.10 ^aA^	2.26 ± 0.42 ^abA^
	D1	5.41 ± 2.51 ^abcA^	0.48 ± 0.10 ^aA^	0.66 ± 0.20 ^aA^	2.03 ± 0.01 ^aA^
	D2	7.03 ± 0.55 ^cA^	0.60 ± 0.01 ^abA^	0.98 ± 0.32 ^aA^	3.57 ± 0.16 ^cdeA^
	D3	4.97 ± 0.83 ^abcA^	0.68 ± 0.07 ^abA^	0.88 ± 0.17 ^aA^	6.62 ± 0.06 ^gB^
	D4	6.04 ± 0.05 ^abcA^	0.75 ± 0.04 ^bA^	0.64 ± 0.17 ^aA^	3.69 ± 0.29 ^cdefB^
	D5	6.66 ± 0.88 ^bcB^	0.60 ± 0.11 ^abA^	1.05 ± 0.25 ^aA^	6.09 ± 0.01 ^gB^
	D6	5.26 ± 0.65 ^abcB^	1.06 ± 0.00 ^cA^	0.75 ± 0.16 ^aA^	4.79 ± 0.32 ^fB^
Haplic Luvisol	Control	3.26 ± 0.17 ^abA^	1.46 ± 0.07 ^deA^	0.71 ± 0.28 ^aA^	3.20 ± 0.08 ^bcdA^
	D1	3.02 ± 0.17 ^aA^	1.36 ± 0.03 ^dA^	0.74 ± 0.19 ^aA^	2.75 ± 0.16 ^abcA^
	D2	3.38 ± 0.31 ^abA^	1.28 ± 0.00 ^cdA^	0.67 ± 0.14 ^aA^	2.64 ± 0.23 ^abcA^
	D3	2.76 ± 0.57 ^aA^	1.66 ± 0.10 ^eB^	0.81 ± 0.01 ^aA^	4.13 ± 0.65 ^defB^
	D4	5.80 ± 0.63 ^abcA^	1.52 ± 0.02 ^deA^	0.81 ± 0.04 ^aA^	3.75 ± 0.32 ^cdefB^
	D5	2.67 ± 0.01 ^aA^	1.35 ± 0.07 ^dA^	0.79 ± 0.01 ^aB^	2.78 ± 0.34 ^abcA^
	D6	4.75 ± 0.38 ^abcB^	1.30 ± 0.12 ^cdA^	0.79 ± 0.02 ^aA^	4.37 ± 0.15 ^efB^
Two-Way ANOVA (*p*-value)					
Type of soil		<0.001	<0.001	0.527	<0.001
Fertilization		0.025	<0.001	0.494	<0.001
Type of soil × Fertilization		0.038	<0.001	0.429	<0.001
2020 (B)					
Gleyic Chernozem	Control	5.63 ± 1.71 ^aA^	0.73 ± 0.01 ^abA^	0.76 ± 0.32 ^aA^	2.18 ± 0.02 ^aA^
	D1	3.37 ± 0.06 ^aA^	0.82 ± 0.13 ^abcdB^	0.96 ± 0.04 ^aA^	3.52 ± 0.22 ^bA^
	D2	3.25 ± 0.87 ^aA^	0.86 ± 0.00 ^abcdB^	0.79 ± 0.05 ^aA^	2.67 ± 0.40 ^abA^
	D3	3.53 ± 0.53 ^aA^	1.00 ± 0.03 ^bcdA^	0.77 ± 0.09 ^aA^	3.10 ± 0.13 ^abA^
	D4	4.01 ± 1.30 ^aA^	0.79 ± 0.05 ^abcdA^	0.88 ± 0.10 ^aA^	3.08 ± 0.23 ^abA^
	D5	2.38 ± 0.08 ^aA^	0.77 ± 0.02 ^abcA^	0.62 ± 0.06 ^aA^	1.93 ± 0.04 ^aA^
	D6	1.85 ± 0.16 ^aA^	0.63 ± 0.12 ^aA^	0.47 ± 0.31 ^aA^	2.13 ± 0.03 ^aA^
Haplic Luvisol	Control	3.34 ± 0.88 ^aA^	0.92 ± 0.15 ^abcAd^	0.79 ± 0.16 ^aA^	2.27 ± 0.41 ^abA^
	D1	3.38 ± 0.97 ^aA^	1.14 ± 0.01 ^dA^	0.55 ± 0.03 ^aA^	2.31 ± 0.52 ^abA^
	D2	2.76 ± 0.33 ^aA^	1.12 ± 0.06 ^cdA^	0.57 ± 0.06 ^aA^	2.63 ± 0.58 ^abA^
	D3	2.34 ± 0.00 ^aA^	0.88 ± 0.02 ^abcdA^	0.57 ± 0.02 ^aA^	2.03 ± 0.59 ^aA^
	D4	3.38 ± 1.44 ^aA^	1.03 ± 0.21 ^bcdA^	0.57 ± 0.09 ^aA^	2.10 ± 0.18 ^aA^
	D5	2.82 ± 0.03 ^aA^	0.97 ± 0.06 ^abcdA^	0.53 ± 0.00 ^aA^	2.03 ± 0.10 ^aA^
	D6	1.92 ± 0.65 ^aA^	1.11 ± 0.06 ^cdA^	0.53 ± 0.04 ^aA^	1.97 ± 0.02 ^aA^
Two-Way ANOVA (*p*-value)					
Type of soil		0.090	<0.001	0.007	0.002
Fertilization		0.018	0.168	0.123	0.002
Type of soil × Fertilization		0.363	0.014	0.220	0.028

Data are expressed as means ± SD. Values in columns marked with the same lowercase letters do not differ significantly at a significance level of 0.05. Values marked with the same capital letters indicate no significant differences between the years of the study at a significance level of 0.05. D1–D6: ash fertilization level.

**Table 4 foods-13-00049-t004:** Thermodynamic characteristics of gelatinization of barley starch pastes.

Type of Soil	Fertilization	T_O_ (°C)	T_P_ (°C)	T_E_ (°C)	ΔT (°C)	ΔH_G_ (J/g)
2019 (A)						
Gleyic Chernozem	Control	60.8 ± 0.7 ^aB^	65.0 ± 0.1 ^abcB^	70.7 ± 0.8 ^aA^	9.9 ± 1.2 ^aA^	9.07 ± 0.31 ^abcdA^
	D1	59.8 ± 0.3 ^aA^	64.7 ± 0.1 ^aA^	70.3 ± 0.3 ^aA^	10.5 ± 0.6 ^aA^	11.38 ± 0.27 ^eB^
	D2	60.7 ± 0.4 ^aB^	64.9 ± 0.2 ^abcB^	70.4 ± 0.3 ^aB^	9.7 ± 0.1 ^aA^	10.22 ± 0.80 ^cdeB^
	D3	60.3 ± 0.7 ^aA^	64.9 ± 0.2 ^abcB^	70.4 ± 0.1 ^aA^	10.1 ± 0.7 ^aA^	10.45 ± 0.22 ^deB^
	D4	61.0 ± 0.4 ^aB^	65.7 ± 0.8 ^cB^	71.5 ± 1.7 ^aA^	10.5 ± 1.3 ^aA^	10.11 ± 0.29 ^cdB^
	D5	60.7 ± 0.0 ^aB^	65.0 ± 0.1 ^abcB^	70.4 ± 0.2 ^aA^	9.7 ± 0.1 ^aA^	8.58 ± 0.52 ^abcA^
	D6	60.7 ± 0.5 ^aB^	65.6 ± 0.4 ^bcB^	71.6 ± 0.7 ^aA^	10.9 ± 1.2 ^aA^	9.32 ± 0.22 ^bcdB^
Haplic Luvisol	Control	60.1 ± 0.0 ^aB^	64.6 ± 0.1 ^aB^	70.1 ± 0.2 ^aB^	9.9 ± 0.2 ^aA^	7.90 ± 0.46 ^abA^
	D1	59.8 ± 0.2 ^aB^	64.8 ± 0.0 ^abB^	70.6 ± 0.2 ^aB^	10.8 ± 0.4 ^aA^	7.86 ± 0.31 ^aA^
	D2	59.6 ± 0.4 ^aB^	64.9 ± 0.2 ^abcB^	70.4 ± 0.3 ^aB^	10.8 ± 0.6 ^aA^	10.22 ± 0.80 ^cdeB^
	D3	59.3 ± 0.2 ^aB^	64.6 ± 0.0 ^aB^	70.1 ± 0.4 ^aA^	10.8 ± 0.6 ^aA^	9.12 ± 0.09 ^bcdB^
	D4	59.3 ± 0.6 ^aB^	64.5 ± 0.1 ^aB^	70.6 ± 0.5 ^aA^	11.3 ± 1.1 ^aA^	7.89 ± 0.14 ^abA^
	D5	59.8 ± 0.5 ^aB^	64.6 ± 0.2 ^aB^	70.2 ± 0.7 ^aA^	10.4 ± 1.2 ^aA^	8.58 ± 0.16 ^abcA^
	D6	60.1 ± 0.5 ^aB^	65.3 ± 0.3 ^abcB^	70.4 ± 1.0 ^aB^	10.3 ± 0.6 ^aA^	8.60 ± 0.44 ^abcB^
Two-Way ANOVA (*p*-value)						
Type of soil		<0.001	0.001	0.058	0.114	<0.001
Fertilization		0.085	0.002	0.253	0.370	<0.001
Type of soil × Fertilization		0.075	0.037	0.448	0.613	<0.000
2020 (B)						
Gleyic Chernozem	Control	58.6 ± 0.4 ^aA^	63.2 ± 0.1 ^aA^	69.1 ± 0.3 ^aA^	10.5 ± 0.7 ^aA^	9.52 ± 0.09 ^eA^
	D1	59.2 ± 0.8 ^aA^	63.5 ± 1.1 ^aA^	69.7 ± 0.9 ^aA^	10.5 ± 0.1 ^aA^	7.64 ± 0.43 ^bcdA^
	D2	58.8 ± 0.2 ^aA^	62.7 ± 0.2 ^aA^	68.1 ± 0.6 ^aA^	9.3 ± 0.2 ^aA^	6.39 ± 1.19 ^abA^
	D3	59.6 ± 1.2 ^aA^	63.3 ± 0.2 ^aA^	69.7 ± 1.2 ^aA^	10.1 ± 0.4 ^aA^	9.04 ± 0.51 ^deA^
	D4	58.6 ± 0.1 ^aA^	62.9 ± 0.1 ^aA^	68.3 ± 0.2 ^aA^	9.7 ± 0.2 ^aA^	8.11 ± 0.56 ^cdeA^
	D5	58.4 ± 0.0 ^aA^	63.2 ± 0.5 ^aA^	68.6 ± 0.7 ^aA^	10.2 ± 0.7 ^aA^	8.90 ± 0.28 ^deA^
	D6	58.1 ± 0.4 ^aA^	62.5 ± 0.7 ^aA^	67.9 ± 1.1 ^aA^	9.8 ± 1.0 ^aA^	6.87 ± 0.61 ^abcA^
Haplic Luvisol	Control	58.4 ± 0.1 ^aA^	63.2 ± 0.2 ^aA^	68.4 ± 0.4 ^aA^	10.1 ± 0.4 ^aA^	7.25 ± 0.15 ^abcA^
	D1	57.8 ± 0.4 ^aA^	62.8 ± 0.3 ^aA^	68.5 ± 0.1 ^aA^	10.7 ± 0.3 ^aA^	6.53 ± 0.60 ^abA^
	D2	57.9 ± 0.2 ^aA^	62.8 ± 0.1 ^aA^	68.4 ± 0.4 ^aA^	10.6 ± 0.3 ^aA^	7.02 ± 0.54 ^abcA^
	D3	58.4 ± 0.2 ^aA^	63.3 ± 0.1 ^aA^	68.9 ± 0.9 ^aA^	10.5 ± 1.0 ^aA^	7.05 ± 0.30 ^abcA^
	D4	57.9 ± 0.3 ^aA^	63.1 ± 0.0 ^aA^	68.6 ± 0.4 ^aA^	10.8 ± 0.7 ^aA^	7.77 ± 0.24 ^bcdA^
	D5	58.2 ± 0.3 ^aA^	63.1 ± 0.0 ^aA^	68.7 ± 0.3 ^aA^	10.5 ± 0.6 ^aA^	6.77 ± 0.20 ^abcA^
	D6	56.5 ± 1.0 ^aA^	61.7 ± 0.2 ^aA^	67.5 ± 1.1 ^aA^	11.1 ± 2.1 ^aA^	5.85 ± 0.10 ^aA^
Two-Way ANOVA (*p*-value)						
Type of soil		<0.001	0.166	0.154	0.021	<0.001
Fertilization		0.001	<0.001	0.016	0.853	<0.001
Type of soil × Fertilization		0.191	0.214	0.409	0.448	<0.001

Data are expressed as means ± SD. Values in columns marked with the same lowercase letters do not differ significantly at a significance level of 0.05. Values marked with the same capital letters indicate no significant differences between the years of the study at a significance level of 0.05. T_O_—onset temperature, T_P_—peak temperature, T_E_—endset temperature, ΔH_G_—enthalpy of gelatinization, ΔH_R_—enthalpy of retrogradation, and R—percentage of retrogradation = (ΔH_R_/ΔH_G_) × 100. D1–D6: ash fertilization level.

**Table 5 foods-13-00049-t005:** Thermodynamic characteristics of retrogradation of barley starch pastes.

Type of Soil	Fertilization	T_P_ (°C)	T_E_ (°C)	ΔT (°C)	ΔH_R_ (J/g)	R (%)
2019 (A)						
Gleyic Chernozem	Control	51.5 ± 2.5 ^abcA^	60.3 ± 1.1 ^abcdB^	18.1 ± 0.7 ^abA^	4.04 ± 0.37 ^abA^	44.5 ± 3.7 ^bA^
	D1	51.5 ± 1.2 ^abcA^	59.9 ± 0.5 ^abcA^	17.4 ± 0.4 ^abA^	3.57 ± 0.10 ^aA^	31.4 ± 1.0 ^aA^
	D2	54.0 ± 0.4 ^abcdeA^	61.8 ± 0.2 ^defB^	18.1 ± 0.6 ^abA^	4.47 ± 0.24 ^bcA^	43.9 ± 3.0 ^bA^
	D3	54.9 ± 0.0 ^bcdeB^	62.9 ± 0.2 ^fgB^	17.3 ± 0.1 ^aA^	4.77 ± 0.03 ^cB^	45.6 ± 1.2 ^bcA^
	D4	55.8 ± 0.2 ^deA^	63.9 ± 0.4 ^gB^	21.8 ± 2.3 ^cdeA^	4.62 ± 0.08 ^bcB^	45.7 ± 0.6 ^bcA^
	D5	56.7 ± 0.2 ^eB^	64.0 ± 0.3 ^gB^	20.3 ± 0.3 ^bcdA^	4.75 ± 0.17 ^cB^	55.4 ± 3.7 ^defB^
	D6	55.4 ± 0.6 ^cdeB^	62.1 ± 0.7 ^efA^	19.8 ± 0.7 ^abcdB^	4.96 ± 0.14 ^cB^	53.3 ± 2.8 ^bcdeA^
Haplic Luvisol	Control	51.0 ± 0.5 ^abA^	61.1 ± 0.5 ^bcdeA^	19.2 ± 1.0 ^abcdA^	4.07 ± 0.18 ^abA^	51.7 ± 4.6 ^bcdeA^
	D1	52.6 ± 0.3 ^abcdA^	60.4 ± 0.1 ^bcdA^	17.9 ± 0.4 ^abA^	4.48 ± 0.09 ^bcB^	57.0 ± 3.4 ^efA^
	D2	52.8 ± 1.2 ^abcdeA^	58.8 ± 0.1 ^aA^	19.0 ± 0.3 ^abcA^	4.54 ± 0.27 ^bcA^	44.5 ± 2.5 ^bA^
	D3	54.9 ± 0.6 ^bcdeA^	61.5 ± 0.5 ^cdefA^	20.3 ± 1.1 ^bcdA^	4.05 ± 0.04 ^abA^	44.4 ± 0.2 ^bA^
	D4	56.6 ± 2.0 ^deA^	63.1 ± 0.1 ^fgA^	22.0 ± 2.0 ^deA^	4.98 ± 0.40 ^cA^	63.1 ± 6.0 ^fA^
	D5	50.0 ± 3.4 ^aA^	59.5 ± 0.9 ^abA^	18.7 ± 0.4 ^abA^	4.69 ± 0.24 ^cA^	54.8 ± 3.6 ^cdefA^
	D6	54.4 ± 0.2 ^bcdeA^	67.0 ± 0.4 ^hB^	24.0 ± 0.2 ^eA^	4.03 ± 0.05 ^abA^	46.9 ± 2.0 ^bcdA^
Two-Way ANOVA (*p*-value)						
Type of Soil		0.015	0.007	0.001	0.452	<0.001
Fertilization		<0.001	<0.001	<0.000	<0.001	<0.001
Type of Soil × Fertilization		0.001	<0.001	0.001	<0.001	<0.001
2020 (B)						
Gleyic Chernozem	Control	47.6 ± 0.8 ^aA^	57.6 ± 0.8 ^aA^	17.7 ± 0.8 ^abcA^	4.16 ± 0.05 ^abcA^	43.7 ± 0.7 ^aA^
	D1	50.4 ± 0.6 ^aA^	59.1 ± 0.4 ^abcA^	17.2 ± 0.4 ^abcA^	3.91 ± 0.16 ^abA^	51.2 ± 2.7 ^abcB^
	D2	46.8 ± 4.7 ^aA^	58.5 ± 0.7 ^abA^	17.9 ± 1.8 ^abcA^	3.95 ± 0.22 ^abA^	62.9 ± 9.7 ^cdeB^
	D3	51.7 ± 0.4 ^aA^	59.7 ± 0.4 ^abcdeA^	17.0 ± 0.4 ^abA^	4.08 ± 0.12 ^abcA^	45.2 ± 1.3 ^abA^
	D4	53.5 ± 27.0 ^aA^	60.7 ± 1.2 ^bcdefA^	17.6 ± 1.2 ^abcA^	3.57 ± 0.11 ^aA^	44.1 ± 3.1 ^aA^
	D5	52.3 ± 0.4 ^aA^	60.7 ± 1.2 ^bcdefA^	17.9 ± 1.8 ^abcA^	3.66 ± 0.16 ^aA^	41.2 ± 2.6 ^aA^
	D6	52.2 ± 0.5 ^aA^	59.5 ± 0.4 ^abcdA^	16.4 ± 0.6 ^aA^	3.51 ± 0.20 ^aA^	51.3 ± 3.9 ^abcA^
Haplic Luvisol	Control	51.4 ± 0.2 ^aA^	61.1 ± 0.3 ^cdefA^	18.3 ± 0.4 ^abcA^	4.08 ± 0.28 ^abcA^	56.4 ± 4.1 ^abcdA^
	D1	52.1 ± 1.1 ^aA^	61.1 ± 0.3 ^cdefA^	18.2 ± 1.0 ^abcA^	4.12 ± 0.14 ^abcA^	63.6 ± 7.3 ^cdeA^
	D2	51.4 ± 0.1 ^aA^	59.7 ± 0.6 ^abcdeA^	18.2 ± 0.9 ^abcA^	4.62 ± 0.09 ^bcdA^	66.1 ± 5.1 ^cdeB^
	D3	55.2 ± 0.1 ^aA^	60.9 ± 0.4 ^cdefA^	19.1 ± 1.1 ^abcA^	4.77 ± 0.08 ^cdB^	67.8 ± 4.0 ^deB^
	D4	55.2 ± 0.5 ^aA^	61.6 ± 0.7 ^defA^	20.2 ± 0.7 ^bcdA^	4.77 ± 0.08 ^cdA^	61.4 ± 2.6 ^bcdeA^
	D5	55.2 ± 0.7 ^aA^	61.8 ± 0.2 ^efA^	20.7 ± 1.0 ^cdA^	5.07 ± 0.08 ^dA^	74.9 ± 2.9 ^efB^
	D6	55.8 ± 1.8 ^aA^	62.3 ± 1.4 ^fA^	23.5 ± 2.5 ^dA^	5.00 ± 0.76 ^dA^	85.4 ± 12.7 ^fB^
Two-Way ANOVA (*p*-value)						
Type of Soil		0.022	<0.001	<0.001	<0.001	<0.001
Fertilization		0.450	<0.001	0.026	0.107	<0.001
Type of Soil × Fertilization		0.492	<0.001	<0.001	<0.001	<0.001

Data are expressed as means ± SD. Values in columns marked with the same lowercase letters do not differ significantly at a significance level of 0.05. Values marked with the same capital letters indicate no significant differences between the years of the study at a significance level of 0.05. D1–D6: ash fertilization level.

**Table 6 foods-13-00049-t006:** Pasting parameters of barley starches.

Type of Soil	Fertilization	PT [°C]	PV [mPa·s]	HPV [mPa·s]	BD [mPa·s]	FV [mPa·s]	SB [mPa·s]
2019 (A)							
Gleyic Chernozem	Control	90.7 ± 0.8 ^aA^	409 ± 2 ^bcB^	336 ± 1 ^cdB^	73 ± 1 ^bcB^	405 ± 2 ^bA^	69 ± 3 ^bA^
	D1	90.4 ± 0.0 ^aA^	424 ± 3 ^cdeB^	347 ± 1 ^deA^	77 ± 2 ^bcdB^	416 ± 4 ^bcA^	69 ± 3 ^bA^
	D2	90.2 ± 0.5 ^aA^	497 ± 11 ^gB^	390 ± 8 ^hB^	106 ± 4 ^eB^	553 ± 11 ^hB^	163 ± 13 ^eB^
	D3	89.6 ± 0.8 ^aA^	412 ± 2 ^cdB^	329 ± 2 ^bcA^	83 ± 1 ^Db^	462 ± 4 ^gA^	133 ± 4 ^dA^
	D4	90.4 ± 0.8 ^aA^	455 ± 14 ^fB^	375 ± 10 ^gB^	80 ± 5 ^bcdB^	457 ± 11 ^fgA^	82 ± 5 ^bcA^
	D5	89.4 ± 0.5 ^a^	391 ± 4 ^bB^	316 ± 4 ^aA^	76 ± 1 ^bcdB^	450 ± 3 ^efgA^	135 ± 2 ^dA^
	D6	90.1 ± 0.5 ^aA^	420 ± 6 ^cdB^	342 ± 4 ^deB^	77 ± 2 ^bcdB^	431 ± 14 ^cdeA^	88 ± 10 ^cA^
Haplic Luvisol	Control	90.6 ± 0.4 a	440 ± 2 ^efA^	363 ± 2 ^fgA^	77 ± 1 ^bcdB^	440 ± 4 ^defA^	77 ± 4 ^bcA^
	D1	89.9 ± 2.2 ^aA^	430 ± 3 ^deB^	349 ± 2 ^deA^	81 ± 2 ^dB^	426 ± 3 ^bcdA^	77 ± 2 ^bcA^
	D2	89.9 ± 0.5 a	413 ± 8 ^cdB^	341 ± 7 ^cdeA^	72 ± 1 ^bB^	410 ± 5 ^bca^	69 ± 3 ^bA^
	D3	90.6 ± 0.4 ^aA^	431 ± 8 ^deB^	351 ± 4 ^efB^	81 ± 4 ^cdB^	444 ± 6 ^defgB^	94 ± 5 ^cB^
	D4	90.7 ± 0.4 ^aA^	415 ± 3 ^cdB^	336 ± 2 ^cdA^	79 ± 3 ^bcdB^	416 ± 7 ^bcA^	80 ± 8 ^bcA^
	D5	90.1 ± 0.9 ^aA^	369 ± 9 ^aA^	315 ± 4 ^aA^	54 ± 5 ^aA^	362 ± 8 ^aA^	47 ± 4 ^aA^
	D6	89.1 ± 0.4 ^aA^	391 ± 4 ^bA^	316 ± 2 ^abA^	75 ± 3 ^bcdB^	383 ± 4 ^aA^	67 ± 5 ^bA^
Two-Way ANOVA (*p*-value)							
Type of Soil		0.962	<0.001	<0.001	<0.001	<0.001	<0.001
Fertilization		0.284	<0.001	<0.001	<0.001	<0.001	<0.001
Type of Soil × Fertilization		0.363	<0.001	<0.001	<0.001	<0.001	<0.001
2020 (B)							
Gleyic Chernozem	Control	92.2 ± 0.9 ^aA^	358 ± 2 ^bcA^	322 ± 1 ^bcA^	36.7 ± 1.5 ^bcdA^	394 ± 7 ^aA^	72 ± 7 ^aB^
	D1	92.0 ± 0.6 ^aB^	393 ± 4 ^ghA^	346 ± 3 ^efA^	47.3 ± 1.2 ^eA^	470 ± 17 ^bcB^	120 ± 12 ^bB^
	D2	91.9 ± 0.7 ^aA^	375 ± 3 ^defA^	341 ± 2 ^eA^	33.7 ± 1.2 ^abA^	459 ± 8 ^bA^	118 ± 7 ^bA^
	D3	92.2 ± 0.4 ^aA^	367 ± 1 ^bcdA^	327 ± 3 ^cdA^	39.7 ± 2.3 ^dA^	560 ± 4 ^fB^	233 ± 6 ^gB^
	D4	91.2 ± 0.7 ^aA^	359 ± 3 ^bcA^	314 ± 3 ^bA^	45.3 ± 0.6 ^eA^	487 ± 4 ^bcA^	173 ± 7 ^defB^
	D5	91.7 ± 0.4 ^aB^	357 ± 2 ^bA^	318 ± 2 ^bcA^	39.0 ± 0.0 ^cdA^	486 ± 3 ^bcB^	167 ± 5 ^cdeB^
	D6	91.9 ± 0.8 ^aB^	332 ± 4 ^aA^	301 ± 3 ^aA^	30.7 ± 1.2 ^aA^	477 ± 7 ^bcB^	175 ± 6 ^efB^
*Haplic Luvisol*	Control	92.4 ± 0.9 ^aA^	408 ± 13 ^iA^	356 ± 9 ^fA^	52.3 ± 4.2 ^fA^	544 ± 18 ^efB^	188 ± 10 ^efB^
	D1	92.1 ± 0.9 ^aA^	386 ± 5 ^efgA^	346 ± 5 ^efA^	40.0 ± 0.0 ^dA^	498 ± 7 ^cdB^	152 ± 2 ^cdB^
	D2	91.6 ± 0.4 ^aA^	389 ± 2 ^fghA^	336 ± 2 ^deA^	52.3 ± 0.6 ^fA^	482 ± 6 ^bcB^	146 ± 7 ^cB^
	D3	91.1 ± 0.7 ^aA^	372 ± 7 ^cdeA^	338 ± 6 ^dea^	34.7 ± 1.5 ^abcA^	412 ± 11 ^aA^	74 ± 8 ^aA^
	D4	90.9 ± 1.1 ^aA^	400 ± 3 ^ghiA^	344 ± 3 ^eB^	56.7 ± 1.5 ^fA^	521 ± 11 ^deB^	177 ± 10 ^efB^
	D5	91.4 ± 0.4 ^aA^	401 ± 5 ^hiB^	346 ± 4 ^efB^	54.7 ± 1.5 ^fA^	538 ± 6 ^efB^	192 ± 3 ^fB^
	D6	91.9 ± 0.0 ^aA^	388 ± 2 ^fghA^	343 ± 2 ^eB^	45.7 ± 0.6 ^eA^	489 ± 4 ^cB^	147 ± 4 ^cB^
Two-Way ANOVA (*p*-value)							
Type of Soil		0.345	<0.001	<0.001	<0.001	<0.001	0.274
Fertilization		0.087	<0.001	<0.001	<0.001	<0.001	<0.001
Type of Soil × Fertilization		0.768	<0.001	<0.001	<0.001	<0.001	<0.001

Data are expressed as means ± SD. Values in columns marked with the same lowercase letters do not differ significantly at a significance level of 0.05. Values marked with the same capital letters indicate no significant differences between the years of the study at a significance level of 0.05. PT—pasting temperature, PV—peak viscosity, HPV—hot paste viscosity, BD—breakdown (PV-HPV), FV—final viscosity, and SB—setback (FV—HPV). D1–D6: ash fertilization level.

**Table 7 foods-13-00049-t007:** Parameters of the power law model and thixotropy hysteresis loop area determined for viscosity curves of barley starch pastes.

Type of Soil	Fertilization	*K* [Pa s*^n^*]	*n* [-]	R^2^	HA [Pa s^−1^]
2019 (A)					
Gleyic Chernozem	Control	3.1 ± 0.1 ^cdB^	0.366 ± 0.010 ^bcdA^	0.984	365 ± 52 ^bA^
	D1	2.9 ± 0.1 ^cA^	0.393 ± 0.010 ^defA^	0.989	328 ± 31 ^bA^
	D2	6.1 ± 0.2 ^gB^	0.335 ± 0.002 ^abA^	0.973	1003 ± 96 ^efA^
	D3	4.8 ± 0.1 ^fB^	0.323 ± 0.003 ^aA^	0.977	723 ± 11 ^cA^
	D4	4.7 ± 0.2 ^fB^	0.373 ± 0.006 ^cdeA^	0.993	1003 ± 66 ^efA^
	D5	4.6 ± 0.2 ^fB^	0.329 ± 0.006 ^aA^	0.991	793 ± 51 ^cdA^
	D6	4.8 ± 0.1 ^fB^	0.346 ± 0.011 ^abcA^	0.990	948 ± 64 ^efA^
Haplic Luvisol	Control	3.9 ± 0.1 ^eA^	0.427 ± 0.009 ^ghA^	0.998	1443 ± 70 ^gA^
	D1	3.8 ± 0.2 ^eB^	0.405 ± 0.014 ^efgA^	0.994	905 ± 34 ^defA^
	D2	2.1 ± 0.1 ^bA^	0.458 ± 0.018 ^hA^	0.994	258 ± 22 ^abA^
	D3	3.5 ± 0.1 ^deB^	0.407 ± 0.005 ^fgA^	0.997	1025 ± 32 ^fB^
	D4	3.7 ± 0.3 ^deB^	0.380 ± 0.004 ^defA^	0.992	864 ± 13 ^cdeA^
	D5	1.2 ± 0.1 ^aA^	0.516 ± 0.017 ^iB^	0.997	148 ± 16 ^aA^
	D6	2.1 ± 0.2 ^bA^	0.435 ± 0.016 ^ghA^	0.991	259 ± 14 ^abA^
Two-Way ANOVA (*p*-value)					
Type of Soil		<0.000	<0.000	-	0.018
Fertilization		<0.000	<0.000	-	<0.000
Type of Soil × Fertilization		<0.000	<0.000	-	<0.000
2020 (B)					
Gleyic Chernozem	Control	2.0 ± 0.1 ^bA^	0.488 ± 0.009 ^deB^	0.996	910 ± 181 ^abcB^
	D1	3.3 ± 0.1 ^fgA^	0.426 ± 0.010 ^abcB^	0.997	1400 ± 67 ^efB^
	D2	2.5 ± 0.2 ^cdeA^	0.475 ± 0.014 ^deB^	0.997	1462 ± 50 ^fB^
	D3	3.8 ± 0.2 ^gA^	0.411 ± 0.008 ^abB^	0.996	1857 ± 94 ^gB^
	D4	2.0 ± 0.2 ^bcA^	0.481 ± 0.019 ^deB^	0.998	1008 ± 48 ^bcdA^
	D5	2.2 ± 0.1 ^bcdA^	0.485 ± 0.012 ^deB^	0.997	1161 ± 59 ^cdeB^
	D6	2.6 ± 0.2 ^cdeA^	0.453 ± 0.008 ^abcdB^	0.998	1433 ± 96 ^efB^
Haplic Luvisol	Control	4.6 ± 0.3 ^hA^	0.406 ± 0.018 ^aA^	0.993	2012 ± 80 ^gB^
	D1	2.6 ± 0.2 ^cdeA^	0.481 ± 0.017 ^deB^	0.994	1391 ± 146 ^eBf^
	D2	2.3 ± 0.2 ^bcdA^	0.471 ± 0.010 ^cdeA^	0.996	638 ± 48 ^aB^
	D3	0.9 ± 0.1 ^aA^	0.616 ± 0.036 ^eB^	0.995	783 ± 46 ^abB^
	D4	2.6 ± 0.1 ^cdeA^	0.477 ± 0.004 ^deB^	0.993	1256 ± 119 ^defB^
	D5	2.9 ± 0.1 ^efB^	0.455 ± 0.010 ^bcdA^	0.992	1344 ± 58 ^efB^
	D6	1.9 ± 0.1 ^bA^	0.519 ± 0.023 ^deB^	0.996	1238 ± 107 ^defB^
Two-Way ANOVA (*p*-value)					
Type of Soil		0.163	<0.000	-	0.010
Fertilization		<0.000	<0.000	-	<0.000
Type of Soil × Fertilization		<0.000	<0.000	-	<0.000

Data are expressed as means ± SD. Values in columns marked with the same lowercase letters do not differ significantly at a significance level of 0.05. Values marked with the same capital letters indicate no significant differences between the years of the study at a significance level of 0.05. *K*—consistency coefficient, *n*—flow behavior index, and HA—hysteresis area. D1–D6: ash fertilization level.

**Table 8 foods-13-00049-t008:** Parameters of the power law equations describing viscoelastic properties (25 °C) of barley starch pastes.

Type of Soil	Fertilization	*K*′ [Pa s*^n^*^′^]	*n*′ [-]	R^2^	*K*″ [Pa s*^n^*^″^]	*n*″ [-]	R^2^
2019 (A)							
Gleyic Chernozem	Control	20.8 ± 0.5 ^abA^	0.113 ± 0.005 ^aA^	0.992	1.5 ± 0.0 ^aA^	0.503 ± 0.008 ^eA^	0.996
	D1	21.1 ± 1.1 ^bA^	0.109 ± 0.007 ^aA^	0.974	1.6 ± 0.1 ^aA^	0.486 ± 0.008 ^deA^	0.997
	D2	31.2 ± 0.2 ^fB^	0.118 ± 0.001 ^aA^	0.990	3.0 ± 0.1 ^gB^	0.407 ± 0.006 ^aA^	0.997
	D3	24.2 ± 0.2 ^deA^	0.120 ± 0.005 ^aA^	0.988	2.1 ± 0.0 ^cdA^	0.426 ± 0.008 ^aB^	0.995
	D4	22.8 ± 1.2 ^bcdA^	0.118 ± 0.07 ^aA^	0.987	1.9 ± 0.1 ^bA^	0.481 ± 0.004 ^cdeB^	0.995
	D5	23.6 ± 0.5 ^cdA^	0.116 ± 0.008 ^aA^	0.993	2.1 ± 0.1 ^bcdA^	0.421 ± 0.008 ^aA^	0.996
	D6	23.1 ± 1.1 ^bcdA^	0.133 ± 0.012 ^aA^	0.993	2.0 ± 0.1 ^bcdA^	0.472 ± 0.013 ^bcdB^	0.993
Haplic Luvisol	Control	21.4 ± 0.7 ^bcA^	0.139 ± 0.015 ^aA^	0.991	1.9 ± 0.1 ^bcA^	0.498 ± 0.004 ^eB^	0.997
	D1	23.8 ± 0.6 ^deA^	0.161 ± 0.019 ^aA^	0.978	2.2 ± 0.1 ^deA^	0.480 ± 0.008 ^bcdBe^	0.995
	D2	22.9 ± 0.8 ^bcdA^	0.152 ± 0.015 ^aA^	0.989	2.1 ± 0.1 ^bcdA^	0.486 ± 0.009 ^deB^	0.989
	D3	24.9 ± 0.5 ^deB^	0.149 ± 0.011 ^aA^	0.988	2.4 ± 0.1 ^efB^	0.462 ± 0.009 ^bcA^	0.995
	D4	26.0 ± 0.6 ^eB^	0.159 ± 0.006 ^aA^	0.990	2.5 ± 0.1 ^fA^	0.457 ± 0.010 ^bA^	0.994
	D5	18.7 ± 0.7 ^aA^	0.154 ± 0.009 ^aA^	0.981	1.5 ± 0.0 ^aA^	0.526 ± 0.007 ^fB^	0.997
	D6	23.9 ± 1.1 ^deB^	0.152 ± 0.008 ^aA^	0.985	2.0 ± 0.1 ^bcdA^	0.487 ± 0.007 ^deB^	0.998
Two-Way ANOVA (*p*-value)							
Type of Soil		0.005	<0.001	-	0.039	<0.001	-
Fertilization		<0.001	0.240	-	<0.001	<0.001	-
Type of Soil × Fertilization		<0.001	0.163	-	<0.001	<0.001	-
2020 (B)							
Gleyic Chernozem	Control	19.4 ± 1.3 ^bA^	0.182 ± 0.011 ^deB^	0.992	2.1 ± 0.1 ^aB^	0.498 ± 0.014 ^eA^	0.995
	D1	25.6 ± 1.1 ^defB^	0.147 ± 0.012 ^abB^	0.992	2.5 ± 0.2 ^bB^	0.462 ± 0.012 ^dA^	0.996
	D2	25.2 ± 1.2 ^defA^	0.150 ± 0.008 ^abcB^	0.993	2.5 ± 0.1 ^bA^	0.465 ± 0.008 ^dB^	0.995
	D3	28.8 ± 0.4 ^gB^	0.136 ± 0.004 ^aB^	0.994	3.3 ± 0.1 ^deB^	0.396 ± 0.004 ^aA^	0.994
	D4	27.9 ± 1.1 ^fgB^	0.156 ± 0.003 ^abcdB^	0.993	3.1 ± 0.1 ^cdeB^	0.428 ± 0.008 ^bcA^	0.996
	D5	24.9 ± 0.7 ^cdeB^	0.162 ± 0.003 ^abcdB^	0.997	3.0 ± 0.1 ^cdB^	0.441 ± 0.008 ^cdB^	0.997
	D6	25.0 ± 1.4 ^deB^	0.154 ± 0.013 ^abcB^	0.993	3.0 ± 0.2 ^cdB^	0.430 ± 0.007 ^cA^	0.997
Haplic Luvisol	Control	29.3 ± 0.4 ^gB^	0.162 ± 0.003 ^abcdB^	0.997	3.7 ± 0.0 ^fB^	0.403 ± 0.005 ^abA^	0.995
	D1	23.1 ± 0.9 ^cdA^	0.176 ± 0.009 ^cdeA^	0.990	3.3 ± 0.1 ^eB^	0.428 ± 0.004 ^bcA^	0.996
	D2	23.0 ± 0.7 ^cdA^	0.183 ± 0.008 ^deB^	0.994	3.4 ± 0.1 ^efB^	0.424 ± 0.005 ^bcA^	0.995
	D3	11.2 ± 0.9 ^aA^	0.194 ± 0.017 ^eB^	0.992	2.0 ± 0.1 ^aA^	0.530 ± 0.011 ^fB^	0.995
	D4	22.3 ± 0.6 ^cA^	0.163 ± 0.003 ^abcdA^	0.993	3.3 ± 0.1 ^deB^	0.423 ± 0.009 ^bcA^	0.995
	D5	26.0 ± 0.1 ^efB^	0.145 ± 0.002 ^abA^	0.991	3.2 ± 0.0 ^deB^	0.423 ± 0.004 ^bcA^	0.995
	D6	19.0 ± 0.6 ^bA^	0.171 ± 0.015 ^bcdeB^	0.991	2.8 ± 0.1 ^bcB^	0.457 ± 0.015 ^dA^	0.991
Two-Way ANOVA (*p*-value)							
Type of Soil		<0.001	<0.001	*-*	<0.001	0.100	*-*
Fertilization		<0.001	0.054	*-*	<0.001	<0.001	*-*
Type of Soil × Fertilization		<0.001	<0.001	*-*	<0.001	<0.001	*-*

Data are expressed as means ± SD. Values in columns marked with the same lowercase letters do not differ significantly at a significance level of 0.05. Values marked with the same capital letters indicate no significant differences between the years of the study at a significance level of 0.05. *K*′, *K*″, *n*′, and *n*″—power low equations constants. D1–D6: ash fertilization level.

## Data Availability

Data is contained within the article.

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
