# Peer review of "Selected Physicochemical, Thermal, and Rheological Properties of Barley Starch Depending on the Type of Soil and Fertilization with Ash from Biomass Combustion"

_foods, 2023, doi:10.3390/foods13010049_

Round 1

Reviewer 1 Report

Comments and Suggestions for Authors

The authors have prepared an interesting manuscript but it is necessary to make certain changes in the manuscript. 

The authors are not precise enough when explaining the obtained results. Also, the authors often use conclusions from other authors to justify the obtained results. Although, the authors mentioned in the chapter results and discussion why is important to know what happens during grain filling period, the authors cannot explain the obtained data exactly. Therefore, it would be good if the authors find more investigations with results from other authors who have conducted similar research on other cereals, such as corn or wheat.

The authors should provide more information about when the experimental plots were fertilized and with what amounts of all fertilizers that were used in the research. For example when fertilization was performed (in the fall or before sowing), how the fertilizers were introduced into the soil and with which kind of equipment. Next, according to which statistical pattern were the experimental plots distributed.

The authors should refine the professional terminology in accordance with the existing scientific literature, (treatments agrotechnical, genetic conditions, replace with applied management, genotype etc)

Comments on the Quality of English Language

The quality of the English language is good and minor changes are needed

Author Response

Reviewer 1

The authors have prepared an interesting manuscript but it is necessary to make certain changes in the manuscript.

Response: The authors of this paper would like to thank you very much for your review. We have made every effort to improve the manuscript according to the reviewers' suggestions.

The authors are not precise enough when explaining the obtained results. Also, the authors often use conclusions from other authors to justify the obtained results. Although, the authors mentioned in the chapter results and discussion why is important to know what happens during grain filling period, the authors cannot explain the obtained data exactly. Therefore, it would be good if the authors find more investigations with results from other authors who have conducted similar research on other cereals, such as corn or wheat.

Response: Thank you for your attention. In our opinion, the research topic undertaken is innovative and there are no scientific works covering the impact of fertilization with alternative ash from biomass combustion on the properties of cereal starches. This topic was discussed in an earlier work in the field of potato starch. However, it is difficult to compare these starches due to their different botanical origins. The introduction to the work additionally describes examples of the impact of fertilization with ashes from biomass combustion on the yield of potatoes and winter barley. In the discussion of the results, work related to the influence of nitrogen fertilization on the gelatinization characteristics of cereal starches was used.

The authors should provide more information about when the experimental plots were fertilized and with what amounts of all fertilizers that were used in the research. For example when fertilization was performed (in the fall or before sowing), how the fertilizers were introduced into the soil and with which kind of equipment. Next, according to which statistical pattern were the experimental plots distributed.

Response: The mineral composition of the ash, the amount of nutrients supplied to the soil each year, and information on when the experimental plots were fertilized and with what amounts of all fertilizers, are given in detail in the paper by Szpunar-Krok et al. [10.            Szpunar-Krok, E.; Szostek, M.; Pawlak, R.; Gorzelany, J.; Migut, D. Effect of fertilisation with ash from biomass combustion on the mechanical properties of potato tubers (Solanum tuberosum L.) grown in two types of soil. Agronomy 2022, 12, 379. https://doi.org/10.3390/ agronomy12020379], which is why it is not described in this paper. In accordance with the Reviewer's suggestion, appropriate additions were made to the text of the manuscript.

The authors should refine the professional terminology in accordance with the existing scientific literature, (treatments agrotechnical, genetic conditions, replace with applied management, genotype etc)

Response: The manuscript has been reviewed. The authors have tried to improve the terminology.

Reviewer 2 Report

Comments and Suggestions for Authors

The study analyzed the impact of fertilizing spring barley with fly ash from biomass combustion grown on two types of soil: Haplic Luvisol (HL) and Gleyic Chernozem (GC). The experiment was conducted in 2019 (A) and 2020 (B), and barley was fertilized with ash doses (D1-D6) differing in the content of minerals, mainly potassium and phosphorus. Starch was isolated from ground samples of barley grain using a laboratory method. In the tested barley starch samples, the amylose content, the clarity of paste and the content of selected minerals were determined using the atomic absorption spectrometry (ASA) method. The thermodynamic characteristics of gelatinization and retrogradation were determined using the DSC method. Pasting characteristics of 10% starch pastes were also characterized using the RVA method, as well as flow curves. The viscoelastic properties of the pastes were also determined using an oscillatory rheometer.

Overall, the current study is innovative and the experimental comparison is conducted with rigor. I recommend to consider the manuscript if the authors can explain the following concerns.

1.     The remaining ingredients are protein 70 (from 11.5-14.2%), β-glucans (3.7-7.7%), fats (4.7-6.8%) and minerals (1.8-2.4%). There should have vary different such as β-glucans. Please refer this reference (Food & Function, 2022, 13(24), 12686-12696.).

2.     “2.2.3. Thermodynamic characteristics of gelatinisation and retrogradation by DSC”

The resistant starch should be measured. Please refer this reference (Comprehensive Reviews in Food Science and Safety, 2023, 22:4217–424.).

3.     Line362-363. The structure, nature of the grains and the physicochemical and rheological properties of starch depend on a number of factors. The The resistant starch should be measured.

4. The reference should be updated in recent years.

Comments on the Quality of English Language

The study analyzed the impact of fertilizing spring barley with fly ash from biomass combustion grown on two types of soil: Haplic Luvisol (HL) and Gleyic Chernozem (GC). The experiment was conducted in 2019 (A) and 2020 (B), and barley was fertilized with ash doses (D1-D6) differing in the content of minerals, mainly potassium and phosphorus. Starch was isolated from ground samples of barley grain using a laboratory method. In the tested barley starch samples, the amylose content, the clarity of paste and the content of selected minerals were determined using the atomic absorption spectrometry (ASA) method. The thermodynamic characteristics of gelatinization and retrogradation were determined using the DSC method. Pasting characteristics of 10% starch pastes were also characterized using the RVA method, as well as flow curves. The viscoelastic properties of the pastes were also determined using an oscillatory rheometer.

Overall, the current study is innovative and the experimental comparison is conducted with rigor. I recommend to consider the manuscript if the authors can explain the following concerns.

1.     The remaining ingredients are protein 70 (from 11.5-14.2%), β-glucans (3.7-7.7%), fats (4.7-6.8%) and minerals (1.8-2.4%). There should have vary different such as β-glucans. Please refer this reference (Food & Function, 2022, 13(24), 12686-12696.).

2.     “2.2.3. Thermodynamic characteristics of gelatinisation and retrogradation by DSC”

The resistant starch should be measured. Please refer this reference (Comprehensive Reviews in Food Science and Safety, 2023, 22:4217–424.).

3.     Line362-363. The structure, nature of the grains and the physicochemical and rheological properties of starch depend on a number of factors. The The resistant starch should be measured.

4. The reference should be updated in recent years.

Author Response

Reviewer 2

The study analyzed the impact of fertilizing spring barley with fly ash from biomass combustion grown on two types of soil: Haplic Luvisol (HL) and Gleyic Chernozem (GC). The experiment was conducted in 2019 (A) and 2020 (B), and barley was fertilized with ash doses (D1-D6) differing in the content of minerals, mainly potassium and phosphorus. Starch was isolated from ground samples of barley grain using a laboratory method. In the tested barley starch samples, the amylose content, the clarity of paste and the content of selected minerals were determined using the atomic absorption spectrometry (ASA) method. The thermodynamic characteristics of gelatinization and retrogradation were determined using the DSC method. Pasting characteristics of 10% starch pastes were also characterized using the RVA method, as well as flow curves. The viscoelastic properties of the pastes were also determined using an oscillatory rheometer.

Overall, the current study is innovative and the experimental comparison is conducted with rigor. I recommend to consider the manuscript if the authors can explain the following concerns.

Response: The authors would like to thank you for reviewing the work. Changes have been made to the manuscript taking into account the Reviewers' suggestions. We hope that it will be correct in its current form.

The remaining ingredients are protein 70 (from 11.5-14.2%), β-glucans (3.7-7.7%), fats (4.7-6.8%) and minerals (1.8-2.4%). There should have vary different such as β-glucans. Please refer this reference (Food & Function, 2022, 13(24), 12686-12696.).

Response: The authors have read the indicated scientific article. It is very interesting and concerns the important problem of hyperlipidemia and how to combat this disease with a diet rich in oatmeal, containing phenolic compounds and β-glucans. These compounds also occur in barley grain and their presence influences the health-promoting properties of this grain. Thank you for your suggestion, the proposed reference has been taken into account in introduction section.

“2.2.3. Thermodynamic characteristics of gelatinisation and retrogradation by DSC”

The resistant starch should be measured. Please refer this reference (Comprehensive Reviews in Food Science and Safety, 2023, 22:4217–424.).

Response: Thank you for this suggestion. As part of the research methodology, starches obtained from the grain being the subject of the experiment were also analyzed using the DSC method, enabling the assessment of changes during gelatinization and retrogradation. The DSC method can primarily assess the retrogradation process of amylopectin. Indeed, retrograded starch is one of the resistant starch fractions, but the aim of our research was slightly different. Thank you for your suggestion, considering barley as a raw material for food production, determining the content of ingredients such as β-glucans and resistant starch will be an interesting direction in the development of our research.

Line362-363. The structure, nature of the grains and the physicochemical and rheological properties of starch depend on a number of factors. The The resistant starch should be measured.

Response: Thank you once again for this suggestion. We agree with the reviewer's opinion that the rheological properties of starch depend on many factors. However, in our opinion, starch resistance is not a key factor here, and the aim of our research did not take into account nutritional aspects and the impact of starch on the human organiam. Therefore, in our research, we used standard, commonly used and available methods to characterize starch. Nevertheless, we would like to thank you for your suggestion, and considering barley as a food ingredient, determining the content of bioactive ingredients, including resistant starch, will be an interesting direction in the development of our further research.

  1. The reference should be updated in recent years.

Response: Other citations were also used in the work. The literature used in the work is current and generally comes from the last ten years. Older literature items concern the analytical methods used in the work. Nevertheless, several important references were added during the revision of the manuscript.

Reviewer 3 Report

Comments and Suggestions for Authors

Dear authors.

I had the pleasure of review your paper which is a valuable work. I only have minor suggestions for your consideration.

Each Figure and table should contain all the required information to the complete understanding without go into the text to look abbreviations and other information.

Dont use italics for the family, and for other names such as Haplic Luvisol.

Delete the margen of the figures.

Comments on the Quality of English Language

The paper is well written 

Author Response

Reviewer 3

Dear authors.

I had the pleasure of review your paper which is a valuable work. I only have minor suggestions for your consideration.

Response: Authors many thanks for your opinion.

Each Figure and table should contain all the required information to the complete understanding without go into the text to look abbreviations and other information.

Response: Information about symbols has been added under tables and charts.

Dont use italics for the family, and for other names such as Haplic Luvisol.

Response: the way some words are written has been improved

Delete the margen of the figures

Response: The figures have been checked. Figures are centrally positioned on the page and are generally of similar size. Stretching them on the page will adversely affect the appearance of the figures.

Reviewer 4 Report

Comments and Suggestions for Authors

The study analyzed the impact of fertilizing fly ash on spring barley starch physicochemical characteristic on two types of soil. There is a lack of innovation and importance of the study. there is also some other suggestions.

1)      The abstract is not concise.

2)      In introduction, most is about common knowledge about barley. There is a lack of progress of ash fertilizer on crop quality.

3)      The introduction of experiment method is rough, it need more detail. What’s soil physical and chemical characteristic? what’s mineral content in ash?

4)      In line 134, what’ unit of “following doses: 0.5 (D2), 1.0 (D3), 1.5 (D4), 2.0 (D5) and 2.5 (D6)”

5)      How to exact starch?

6)      RVA is more related with total starch content, why the author only present amylose content? What’s the content of total starch,

7)      Some data of mineral content is wrong, for example P and K, please check.

8)      Mineral usually lie in near pericarp, why did author want to measure them in starch?

Comments on the Quality of English Language

the english can be improved.

Author Response

Reviewer 4

The study analyzed the impact of fertilizing fly ash on spring barley starch physicochemical characteristic on two types of soil. There is a lack of innovation and importance of the study. there is also some other suggestions.

Response: We would like to thank the reviewer for this opinion. In our opinion, the concept of the work is, after all, innovative because it concerns a not yet described issue related to the use of ash from biomass combustion as an alternative fertilizer. An attempt was made to explain the influence of such fertilization and the type of soil on the properties of barley starch. The research was conducted on material from two years of cultivation. We believe that the presented research results are interesting and will expand knowledge in this field.

1)      The abstract is not concise.

Response: The abstract was corrected and shortened according to the Reviewer's and Editor's instructions. The abstract is slightly more than 200 words long, but some information cannot be omitted.

2)      In introduction, most is about common knowledge about barley. There is a lack of progress of ash fertilizer on crop quality.

Response: Thank you for your attention. In our opinion, the research topic undertaken is innovative and there are no scientific works covering the impact of fertilization with alternative ash from biomass combustion on the properties of cereal starches. This topic was discussed in an earlier work in the field of potato starch. The introduction to the work additionally describes examples of the impact of fertilization with ashes from biomass combustion on the yield of potatoes and winter barley.

3)      The introduction of experiment method is rough, it need more detail. What’s soil physical and chemical characteristic? what’s mineral content in ash?

Response: The mineral composition of the ash, the amount of nutrients supplied to the soil each year, and information on when the experimental plots were fertilized and with what amounts of all fertilizers, are given in detail in the paper by Szpunar-Krok et al. [10.            Szpunar-Krok, E.; Szostek, M.; Pawlak, R.; Gorzelany, J.; Migut, D. Effect of fertilisation with ash from biomass combustion on the mechanical properties of potato tubers (Solanum tuberosum L.) grown in two types of soil. Agronomy 2022, 12, 379. https://doi.org/10.3390/ agronomy12020379], which is why it is not described in this paper. In accordance with the Reviewer's suggestion, appropriate additions were made to the text of the manuscript.

4)      In line 134, what’ unit of “following doses: 0.5 (D2), 1.0 (D3), 1.5 (D4), 2.0 (D5) and 2.5 (D6)”

Response: These are biomass ash doses in t·ha-1.

5)      How to exact starch?

Response: Starch was isolated from barley grain using a laboratory method dedicated to cereal starches. The method involved wet rinsing the starch and centrifuging it to remove water and possible impurities. The additional informations were added to the materials section.

6)      RVA is more related with total starch content, why the author only present amylose content? What’s the content of total starch,

Response: The aim of the study was to assess the impact of fertilization with ash from biomass combustion and soil type on selected physicochemical properties of barley starch. Therefore, the study did not examine the content of this biopolymer itself and the influence of the tested factors on this parameter. However, thank you for this attention. We will consider this when planning further research. The content of amylose and its ratio to the content of amylopectin have an extremely important impact on the rheological properties of starch pastes. Therefore, in all tests related to the characterization of starch, the content of amylose, i.e. the linear fraction of starch, must be determined.

7)      Some data of mineral content is wrong, for example P and K, please check.

Response: Thank you for your comment. Indeed, the results of P content in starch were incorrect. The P content values have been corrected in Table 2. The results for K concentration in starch have been checked and are correct. Probably the K concentration in starch is low, due to the method used to obtain starch from barley grains. Washing barley grains several times most likely resulted in the washing out of K, which is easily soluble.

8)      Mineral usually lie in near pericarp, why did author want to measure them in starch?

Response: The influence of various factors (fertilization, soil type) on selected physicochemical properties of starch was examined in the trowel. In the case of starch of various botanical origins, the content and type of minerals influence its physicochemical and rheological properties (gelatinization characteristics, gel texture). Hence the need to determine micro and macro elements.

Round 2

Reviewer 2 Report

Comments and Suggestions for Authors

It can be accepted in the current revision.

Comments on the Quality of English Language

It can be accepted in the current revision.